# Removing Strong Attribute Bias from Neural Networks with Adversarial Filtering

## Abstract

Ensuring a neural network is not relying on protected attributes (*e.g.*, race, sex, age) for prediction is crucial in advancing fair and trustworthy AI. While several promising methods for removing attribute bias in neural networks have been proposed, their limitations remain under-explored. To that end, in this work, we mathematically and empirically reveal the limitation of existing attribute bias removal methods in the presence of strong bias and propose a new method that can mitigate this limitation. Specifically, we first derive a general non-vacuous information-theoretical upper bound on the performance of any attribute bias removal method in terms of the bias strength, revealing that they are effective only when the inherent bias in the dataset is relatively weak. Inspired by this theoretical finding, we then propose a new method using an adversarial objective that directly filters out protected attributes in the input space while maximally preserving all other attributes, without requiring any specific target label. The proposed method achieves state-of-the-art performance in both strong and moderate bias settings. We provide extensive experiments on synthetic, image, and census datasets, to verify the derived theoretical bound and its consequences in practice, and evaluate the effectiveness of the proposed method in removing strong attribute bias.

## 1 Introduction

*Protected attributes* is a term originating from Sociology (Ore & Kurtz, 2000) referring to a finite set of attributes that must not be used in decision-making to prevent exacerbating societal biases against specific demographic groups (Corbett-Davies & Goel, 2018). For example, in deciding whether or not someone should be qualified for a bank loan, race (as one of the protected attributes) must not influence the decision. Given the widespread use of neural networks in real-world decision-making, developing methods capable of explicitly excluding protected attributes from the decision process – more generally referred to as removing attribute bias (Stone et al., 2022) – is of paramount importance.

While many methods for removing attribute bias in neural networks have been proposed (Alvi et al., 2018; Kim et al., 2019; Wang et al., 2020; Nam et al., 2020; Tartaglione et al., 2021; Zhu et al., 2021; Hong & Yang, 2021), the limitations of these methods remain under-explored. In particular, existing studies explore the performance of attribute bias removal methods only in cases where the protected attribute (*e.g.*, race) is *not strongly predictive* of the prediction target (*e.g.*, credit worthiness). However, this implicit assumption does not always hold in practice, especially in cases where training data is scarce. For example, in diagnosing Human Immunodeficiency Virus (HIV) from Magnetic Resonance Imaging (MRI), HIV-positive subjects were found to be significantly older than control subjects, making *age* (a protected attribute) a strong predictor of HIV (Adeli et al., 2021). Another example is the Pima Indians Diabetes Database which contains only 768 samples where several spurious attributes become strongly associated with diabetes diagnosis (Smith et al., 1988; Li & AbdAlmageed, 2024). Even the widely-used CelebA dataset (Liu et al., 2015) contains strong attribute biases. For example, in predicting hair color, sex is a strong predictor[1]. Therefore, it is crucial to study attribute bias removal methods beyond the moderate bias setting to better understand their limitations and the necessary conditions for their effectiveness.

---

[1]See Appendix 2 for detailed attribute bias statistics in real-world datasets.

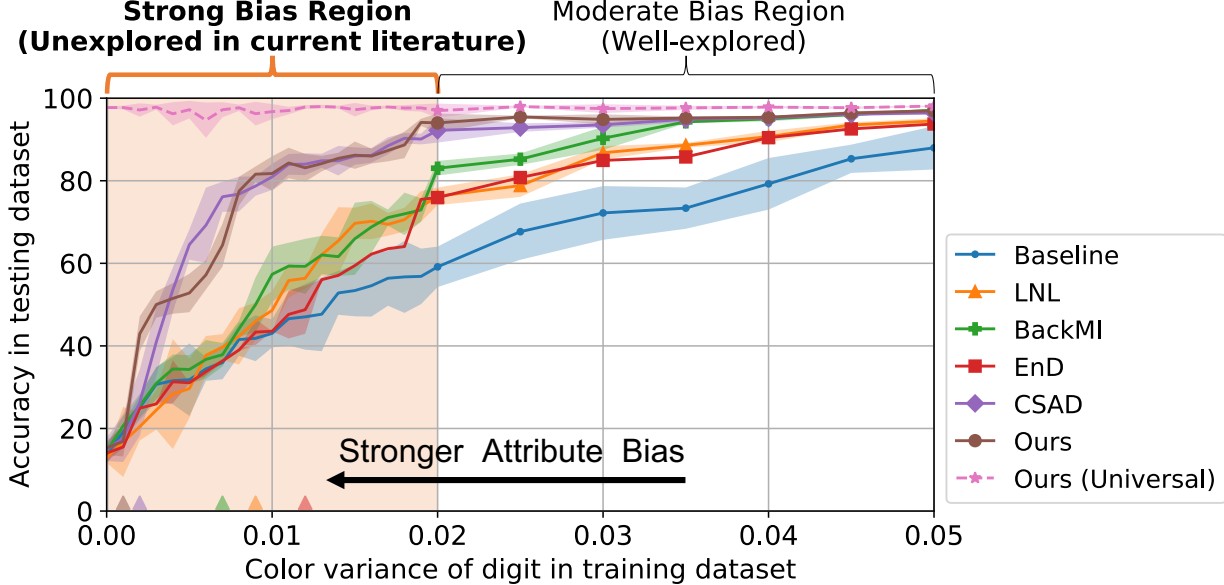

Figure 1: Digit prediction accuracy of bias removal methods trained under different levels of color bias strength in Colored MNIST, showing results on the unexplored region of color variance < 0.02. The breaking point of each method, where its performance becomes statistically similar to the baseline classifier, is labeled with ▲ on the x-axis. While all methods clearly outperform the baseline in the moderate bias region, their effectiveness sharply declines towards the baseline as the bias strength increases. Our proposed method shows a lower breaking point, and no breaking point when a universal distribution is available. The plot shows average accuracy (lines) with one standard deviation error (shaded) over 15 randomized training runs. Further details are provided in Appendix 6.

In Fig. 1, we utilize a specific example to illustrate the limitation in attribute bias removal methods that we will later extensively investigate, mathematically and empirically, in this work. In this example, we conduct an extended version of a popular controlled experiment for evaluating the performance of attribute bias removal methods (Kim et al., 2019; Zhu et al., 2021; Ragonesi et al., 2021). The task is to predict digits from colored MNIST images (Kim et al., 2019) where color is considered a protected attribute. During training, each digit is assigned a unique RGB color with a variance (*i.e.*, the smaller the color variance, the more predictive the color is of the digit, and the stronger the attribute bias). To measure how much the trained model relies on the color (protected attribute) for predicting the digit, model accuracy is reported on a held-out subset of MNIST with a uniformly random color-to-digit assignment (*i.e.*, where the color is not predictive of the digit). While state-of-the-art methods (Kim et al., 2019; Tartaglione et al., 2021; Zhu et al., 2021; Ragonesi et al., 2021) report results for the color variance only in the range [0.02, 0.05] (without providing any justification for this particular range), we explore results for the missing range of [0, 0.02], which we denote as *strong bias region*. In Fig. 1, we observe that the effectiveness of all existing methods sharply declines in the strong bias region, and there exists a *breaking point* in their effectiveness. The breaking point of a bias removal method is defined as the weakest bias strength at which its performance becomes indistinguishable[2] from the baseline classifier that has no bias removal mechanism. The main goal of this paper is to study the cause and extent of this limitation mathematically and empirically. We summarize our main contributions below:[3]

---

[2]Indistinguishable under a two-sample one-way Kolmogorov-Smirnov test with a significance level of 0.05.

[3]This work is an extended version of our paper (Li et al., 2023) presented in the Algorithmic Fairness through the Lens of Time Workshop at NeurIPS 2023. In this work, we further derive a necessary condition for the existence of any method that can remove attribute bias, propose a new method for strong attribute bias removal, and analyze its performance extensively.

- Deriving and verifying a non-vacuous information-theoretic upper bound for the performance of any attribute bias removal method, thereby formalizing the cause and extent of their limitations (Sec. 3).

- Constructing a new method for strong attribute bias removal based on the theoretic finding (Sec. 5).

- Providing an extensive empirical analysis of the proposed method in both moderate and strong bias settings, demonstrating its state-of-the-art performance (Sec. 6).

**In contrast to the state-of-the-art bias-removal methods reviewed in Sec. 2, our method is: 1) target-agnostic** (whereas existing methods need both the downstream prediction target and attribute labels to remove bias), **2) removing bias directly in the input space** (whereas existing methods try to learn an unbiased latent representation), and **3) a simple data pre-processing for downstream tasks** (whereas existing methods need to modify the downstream neural network architecture and its training objective).

## 2 Related Work

**Bias in Neural Networks.** Mitigating bias and improving fairness in neural networks has received considerable attention in recent years (Hardt et al., 2016; Calders et al., 2009; Kusner et al., 2017; Dwork et al., 2012; Chen et al., 2019; Li & Abd-Almageed, 2021; Roh et al., 2020; Kamishima et al., 2011; Cho et al., 2020; Ghassami et al., 2018; Cheng et al., 2021). The methods proposed for mitigating bias in neural networks can be broadly grouped into two categories: 1) methods that aim to mitigate the uneven performance of neural networks between majority and minority groups; and 2) methods that aim to reduce the dependence of neural network prediction on specific attributes. Most notable examples of the former group are methods for constructing balanced training set (Buolamwini & Gebru, 2018; Karkkainen & Joo, 2021), synthesizing additional samples from the minority group (Balakrishnan et al., 2021; Li & Abd-Almageed, 2023), importance weighting the under-represented samples (Wang & Deng, 2020), and domain adaptation techniques that adapt well-learnt representations from the majority group to the minority group (Wang et al., 2019; Guo et al., 2020; Kan et al., 2015). In this work, we focus on the second aim of removing attribute bias from prediction. Existing attribute-removal methods minimize the loss of target prediction from a learnable latent representation while minimizing the mutual information (MI) between the latent representation and protected attributes, either *explicitly* or *implicitly*.

**Explicit Mutual Information Minimization.** These methods mainly differ in the way they estimate MI between latent features and protected attributes, which is then directly minimized together with the classification loss. Most notably, **LNL** (Kim et al., 2019) estimates MI using an auxiliary distribution, **BackMI** (Ragonesi et al., 2021) uses a neural estimator (Belghazi et al., 2018), and, **CSAD** (Zhu et al., 2021) minimizes MI between a latent representation to predict target and another latent representation to predict the protected attributes(Hjelm et al., 2018).

**Implicit Mutual Information Minimization.** Another group of methods aims to remove attribute bias by constructing surrogate losses that implicitly reduce the mutual information between protected attributes and learnt features. Most notably, **LfF** (Nam et al., 2020) proposes training two models simultaneously, where the first model will prioritize easy features for classification by amplifying the gradient of cross-entropy loss with the predictive confidence (softmax score), and the second model will down-weight the importance of samples that are confidently classified by the first model, thereby discouraging predictive features that are easy-to-learn, which are in turn likely to be spurious features with large MI with protected attributes; **EnD** (Tartaglione et al., 2021) adds regularization terms to the target classification (cross-entropy) loss to push apart the feature vectors of samples with the same protected attribute label; **BlindEye** (Alvi et al., 2018) minimizes the target classification loss, as well as the cross-entropy between the uniform distribution and the prediction of a protected attribute classifier operating on the latent features, so that the shared feature vector is not predictive of the protected attribute; **DI** (Wang et al., 2020) learns a shared representation with an ensemble of separate classifiers per domain (*i.e.*, a group of samples having the same protected attribute) to ensure that the prediction from the ensemble model is not biased towards any one domain; **BCL** (Hong & Yang, 2021) proposes Bias-Contrastive loss, which regularizes the feature space by bringing samples of the same target label but different protected attribute label closer; **Group DRO** (Sagawa

et al., 2019) minimizes classification performance gap across groups of samples with different values of the protected attribute by mapping data to a space where the different group distributions are indistinguishable while retaining task-relevant information within each group; **EIIL** (Creager et al., 2021) proposes a two-stage method that initially infers domain partitions and then employs invariant learning (Ganin et al., 2016; Zhao et al., 2019; Albuquerque et al., 2019; Ahmed et al., 2020) to learn features that remain consistent across groups that have different values of the protected attribute; **JTT** (Liu et al., 2021) begins by training a standard Empirical Risk Minimization (ERM) model to identify misclassified examples and then trains a second model to up-weight these examples; and, **CNC** (Zhang et al., 2022) uses a trained ERM model to detect samples with the same target label but dissimilar protected attribute label and trains a new model with contrastive learning to align representations for these samples.

**Generative Dataset Augmentation.** A recent group of methods (Sauer & Geiger, 2021; Goel et al., 2020; Kim et al., 2021; Ramaswamy et al., 2021) aims to mitigate attribute bias by generating counterfactual synthetic samples that can augment the original biased training set to reduce its inherent bias strength. These methods use generative models (*e.g.*, Generative Adversarial Networks (Goodfellow et al., 2014)) to synthesize images of a given biased dataset by randomly altering the protected attribute, a technique commonly denoted *attribute flipping*. Compared with MI-based methods, these generative models address attribute bias by constructing a semi-synthetic dataset with reduced bias strength rather than minimizing mutual information between learned features and protected attributes. Most notably, **CAMEL** (Goel et al., 2020) starts by employing a CycleGAN (Zhu et al., 2017) to learn the semantic transformations between latent features with the same target attribute but different protected attribute, and then performs data augmentations by manipulating the latent features for classifier training; **BiaSwap** (Kim et al., 2021) first employs a biased classifier to divide samples into bias-guiding and bias-contrary categories, and then incorporates the style-transferring module of the image translation model to produce bias-swapped images which retain bias-irrelevant features from bias-guiding samples while inheriting protected attributes from bias-contrary samples; **GAN-Debiasing** (Ramaswamy et al., 2021) formulates two hyperplanes to represent both the target attribute and the protected attribute, and generates synthetic images that retain the appearance of the target attribute while flipping the protected attribute by perturbing latent vector in the protected attribute hyperplane; and, **CGN** (Sauer & Geiger, 2021) learns three predefined independent mechanisms for shape, texture, and background based on domain knowledge, and leverages them to generate images with desired attributes.

**Trade-offs between Bias Removal and Model Utility.** The trade-offs between fairness and accuracy in machine learning models have garnered significant discussion. Most notably, Kleinberg *et al.* (Kleinberg et al., 2016) prove that except in highly constrained cases, no method can simultaneously satisfy three fairness conditions for prediction: *calibration within groups*, *balance for the negative class*, and *balance for the positive class*; and, Dutta *et al.* (Dutta et al., 2020) theoretically demonstrate that, under certain conditions, it is possible to simultaneously achieve optimal accuracy and fairness in terms of *equal opportunity* (Hardt et al., 2016) which requires even false negative rates or even true positive rates across groups. Different from the fairness criteria discussed in these works, we focus on another well-known fairness criterion, *demographic parity* (Kusner et al., 2017; Dwork et al., 2012), which requires even prediction probability across groups, *i.e.*, independence between model prediction and protected attributes. Regarding this criterion, Zhao and Gordon (Zhao & Gordon, 2022) show that any method designed to learn fair representations, while ensuring model predictions are independent of protected attributes, faces an information-theoretic lower bound on the joint error across groups. In contrast, we derive a general information-theoretic upper bound on the best attainable performance, which is not limited to the case where model predictions are independent of protected attributes and considers different levels of the retained protected attribute information in the learnt features.

## 3 Information-Theoretic Bounds on the Performance of Attribute Bias Removal

The observations in Fig. 1 reveal that the existing methods are not effective when the attribute bias is very strong, *i.e.*, all methods have a breaking point, and that there is a negative correlation between their effectiveness and the strength of the attribute bias. However, so far, these observations are limited to the particular Colored MNIST dataset. In this section, we show that this phenomenon is in fact much more

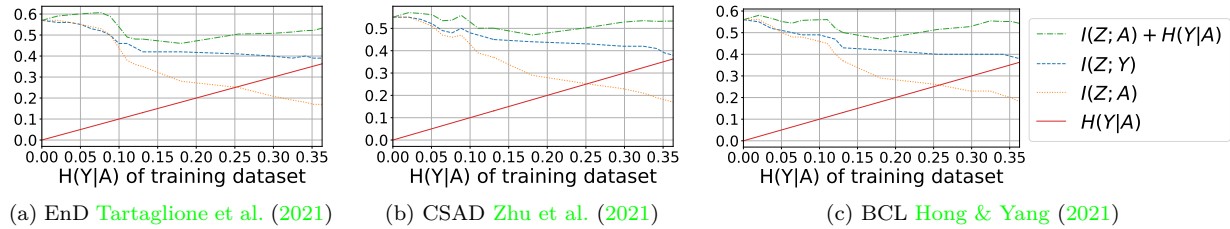

(a) EnD Tartaglione et al. (2021)   (b) CSAD Zhu et al. (2021)   (c) BCL Hong & Yang (2021)

Figure 2: Empirical verification of the bound in Theorem 1 across several bias removal methods trained on real-world datasets (*e.g.*, CelebA), with proof provided in Appendix 1, The x-axis shows $H(Y|A)$, which we vary directly by adjusting the fraction of bias-conflicting images while ensuring a constant number of biased images in the training set. We empirically compute $H(Y|A)$ based on the distribution of $Y$ and $A$ in the modified training set, and estimate mutual information using Belghazi et al. (2018). The bound $0 \leq I(Z;Y) \leq I(Z;A) + H(Y|A)$ holds for all methods (results of additional bias removal methods are provided in Appendix 4).

general. We will elucidate the cause and extent of the limitations we observed in Fig. 1 by deriving a domain-agnostic and data-independent upper bound on the classification performance of any attribute bias removal method in terms of the bias strength.

We first formalize the notions of performance, attribute bias strength, and attribute bias removal. Let $X$ be a random variable representing the input (*e.g.*, images or credit score) with support $\mathcal{X}$, $Y$ be a random variable representing the prediction target (*e.g.*, hair color or credit worthiness) with support $\mathcal{Y}$, and $A$ be a random variable representing the protected attribute (*e.g.*, sex or race). We define the attribute bias removal method as a function $f : \mathcal{X} \to \mathcal{Z}$ that maps input data to a latent bottleneck feature space $\mathcal{Z}$ inducing the random variable $Z$, and consider the prediction model as a function $g : \mathcal{Z} \to \mathcal{Y}$ inducing the random variable $\hat{Y}$. According to the information bottleneck theory (Tishby & Zaslavsky, 2015; Shwartz-Ziv & Tishby, 2017), the goal of classification can be stated as maximizing the mutual information between prediction and target, namely $I(\hat{Y};Y)$, which is itself bounded by the mutual information between the feature and the target due to the data processing inequality (Cover & Thomas, 2006), *i.e.*, $I(\hat{Y};Y) \leq I(Z;Y)$. Intuitively, $I(Z;Y)$ measures how informative the features learnt by the model are of the target, with $I(Z;Y) = 0$ indicating completely uninformative learnt features; *i.e.*, the best attainable prediction performance is no better than random guess. Therefore, the optimization objective of attribute bias removal methods can be formalized as learning $f$ parameterized by $\theta$ that minimizes mutual information between the feature and the protected attribute $I(Z_\theta;A)$, while maximizing mutual information between the feature and the target $I(Z_\theta;Y)$, where $Z_\theta = f_\theta(X)$.

Given the above definitions, we can state our goal concretely: to derive a connection between $H(Y|A)$ (the attribute bias strength measured by the conditional entropy of the target given the protected attribute), $I(Z;A)$ (the amount of the remained attribute bias in the learnt feature) and $I(Z;Y)$ (the best attainable performance on predicting the target from the learnt feature). Note that smaller $H(Y|A)$ corresponds to stronger attribute bias (*i.e.*, the protected attribute can more certainly predict the target). **We first consider the extreme attribute bias ($H(Y|A) = 0$) setting, in which we show that no classifier can outperform random guess if the protected attribute is removed from the learnt feature.**

**Proposition 1.** *Given random variables $Z, Y, A$, in case of the extreme attribute bias,* i.e., $H(Y|A) = 0$, *if the protected attribute is removed from the feature,* i.e., $I(Z;A) = 0$, *then no classifier can outperform random guess,* i.e., $I(Z;Y) = 0$.[4]

This proposition explains and extends the observation on the leftmost location of the x-axis in Fig. 1: when the color variance is zero, color is completely predictive of the digit, *i.e.*, $H(Y|A) = 0$, and removing color from the latent feature, *i.e.*, $I(Z;A) = 0$, makes the prediction uninformative, *i.e.*, $I(Z;Y) = 0$. However, Proposition 1 does not explain the performance beyond just the zero color variance. **To explain the**

---

[4]Proof in Appendix 1.

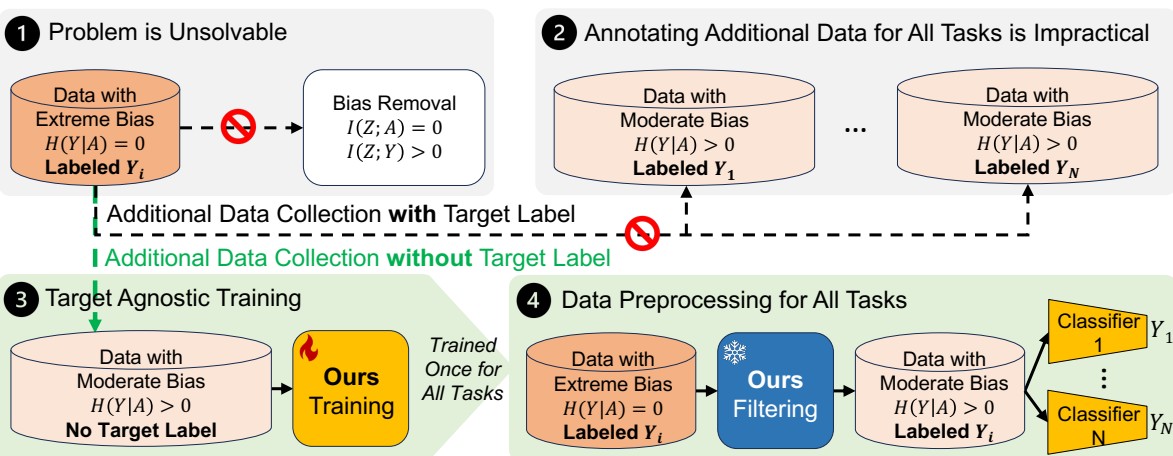

Figure 3: Illustration of extreme bias and the proposed method. (1) In extreme bias where $H(Y|A) = 0$, no effective attribute bias removal method exists unless it can access a universal distribution where $H(Y|A) > 0$. (2) It is impractical to collect samples from a universal distribution with target labels for all potential downstream tasks. (3) Thus, we propose a target-agnostic method that can utilize a universal distribution without target labels, *i.e.*, a partially-observable distribution. (4) Due to its same-space design, our method can be easily applied as preprocessing in various downstream tasks for removing attribute bias.

**performance beyond just the extreme bias setting, the following theorem provides a bound on the performance of attribute bias removal methods in terms of the attribute bias strength**, thus providing a more complete picture of the limitations of such methods and elucidating the connection between performance and bias strength.

**Theorem 1.** *Given random variables $Z, Y, A$, the following inequality holds without exception:*[4]

$$0 \leq I(Z; Y) \leq I(Z; A) + H(Y|A) \tag{1}$$

**Remark 1.** *In the extreme bias case $H(Y|A) = 0$, the bound in Eq. (1) shows that the model performance is bounded by the amount of protected attribute information that is retained in the feature, namely $I(Z; Y) \leq I(Z; A)$. This puts the model in a trade-off: the more the attribute bias is removed, the lower the best attainable performance.*

**Remark 2.** *When the protected attribute is successfully removed from the feature $I(Z; A) = 0$, the bound in Eq. (1) shows that the model's performance is bounded by the strength of the attribute bias, namely $I(Z; Y) \leq H(Y|A)$. This explains the gradual decline observed in Fig. 1 as we move from the moderate to the strong bias region (right to left).*

**Remark 3.** *When $H(Y|A) = 0$ and $I(Z; A) = 0$, Eq. (1) reduces to Proposition 1, $I(Z; Y) = 0$, hence no classifier can outperform random guess.*

**Remark 4.** *Note that the bound is placed on the best attainable performance. So decreasing the bound will decrease performance, but increasing the bound only relaxes the limit without guaranteeing any improvement. For example, consider the baseline classifier: even though there is no attribute bias removal performed (therefore $I(Z; A) \gg 0$), the model declines in the strong bias region since learning the highly predictive protected attribute is very likely in the non-convex optimization.*

To empirically validate Theorem 1 on real-world data, we compute its terms for several existing methods on CelebA and plot the results in Fig. 2. In these experiments, hair color is the target $Y$, and sex is the protected attribute $A$. We vary the bias strength $H(Y|A)$ by increasing/decreasing the fraction of bias-conflicting images in the training set (*i.e.*, images of females with non-blond hair and males with blond hair) while maintaining the number of biased images in training set at 89754. Then, we compute $H(Y|A)$ directly and estimate the mutual information terms $I(Z; A)$ and $I(Z; Y)$ using mutual information neural

estimator (Belghazi et al., 2018). We observe that the bound holds in accordance with Theorem 1 for all methods.

We further investigate the extent of the consequences of the bound for attribute bias removal methods in real-world image and census datasets in Secs. 6.1 and 6.2. In the following section, we investigate whether methods could be designed to mitigate the limitation of removing strong bias.

## 4 Necessary Condition to Remove Extreme Bias

According to Theorem 1, if a dataset has extreme bias ($H(Y|A) = 0$), then the best attainable performance of any attribute bias removal method in learning the latent feature $Z_\theta$ becomes bounded by the amount of attribute bias that remains in the learnt latent feature, $i.e.$, $I(Z_\theta; Y) \leq I(Z_\theta; A)$. Therefore, the more attribute bias the method removes, the lower the best attainable performance on predicting the target from learnt feature becomes. Given that the trade-off is inevitable when there only exists a dataset characterized by extreme bias ($H(Y|A) = 0$), the possibility of sidestepping this trade-off arises only if there exists another dataset following specific distribution ($H(Y|A) > 0$). In this section, we formally derive a necessary condition regarding this possibility. Consider again the random variables $X$ (input), $Y$ (target), and $A$ (protected attribute), as defined in Sec. 3, with the respective distributions $p_X(x)$, $p_Y(y)$, and $p_A(a)$. Note that while the observed joint distribution $p(x, y, a)$ over these random variables in a given dataset can be such that $H_p(Y|A) = 0$, $i.e.$, having extreme bias, this is not necessarily the only observable joint distribution over these random variables. In other words, there could exist another joint distribution $q(x, y, a)$ over the same three random variables (with the correct marginal distributions) in which $H_q(Y|A) > 0$, which we denote as the *universal distribution*. If such a distribution exists – even if yielding no target labels – it could help mitigate the limitation in removing extreme bias in the collected dataset. The following corollary of Theorem 1 shows that the existence of a universal distribution is necessary for the existence of a successful attribute bias removal method.

**Definition 1.** *(Universal Distribution). $q : \mathcal{X} \times \mathcal{Y} \times \mathcal{A} \rightarrow \mathbb{R}^{\geq 0}$ is a universal distribution if all the following conditions hold:*

1. $\sum_{x,y,a} q(x, y, a) = 1$

2. $\sum_{y,a} q(x, y, a) = p_X(x)$

3. $\sum_{x,a} q(x, y, a) = p_Y(y)$

4. $\sum_{x,y} q(x, y, a) = p_A(a)$

5. $H_q(Y|A) > 0$

**Corollary 1.** *(Necessary Condition). Consider any family of bias removal methods $\Theta$, then there exists a method $\phi \in \Theta$ that simultaneously removes the bias and achieves the best performance,* i.e.*, $\phi = \arg\min_{\theta \in \Theta} I(Z_\theta; A) = \arg\max_{\theta \in \Theta} I(Z_\theta; Y)$ only if $\exists\, q(x, y, a) : H_q(Y|A) > 0$.*[4]

The existence of a universal distribution is essentially formalizing the knowledge that the two concepts $A$ and $Y$ are not exactly the same, $i.e.$, there exists a distribution where they can be distinguished. However, note that Corollary 1 does not require this distribution to yield both target labels $Y$ and protected attribute labels $A$ in order to break the trade-off between performance and bias removal. Therefore, assuming universal distribution exists and we can collect samples of input $X$ from it, we consider three possibilities regarding the observability of target $Y$ and protected attribute $A$: 1) **Fully-Observable** where both target and protected attribute labels can be collected; 2) **Partially-Observable** where target labels cannot be collected; and 3) **Non-Observable** where neither target nor protected attribute labels can be collected.

In practice, as verified in Sec. 6, samples of $X$ from a universal distribution can be obtained from large-scale web-scraped datasets or pretrained generative models. However, collecting target labels for numerous downstream tasks is prohibitively expensive due to limited access to subject-matter experts and annotation costs. In contrast, collecting protected attribute labels is more feasible since there are only a small number

of protected attributes, and once the labels are collected, they can be used with any downstream task[5]. **This motivates the development of attribute bias removal methods that do not require target labels. Note that existing SOTA methods cannot utilize the dataset collected from partially-observable or non-observable universal distribution since their training requires target labels.** We construct methods that can utilize a partially-observable universal distribution in Sec. 5, and methods that can utilize a non-observable universal distribution in Sec. 7.

## 5 Bias Removal Using Adversarial Filtering

In this section, we explore how to address the challenge of strong bias as detailed in the previous sections. We saw in Theorem 1 that when the training dataset exhibits extreme bias $H(Y|A) = 0$, removing attribute bias with respect to $A$ is unsolvable for any model. Intuitively, if we live in a hypothetic scenario where the protected attribute $A$ and the target $Y$ are exactly the identical concept $H(Y|A) = 0$, we cannot retain all information of $Y$ while simultaneously disregarding the information of $A$ since $A$ is inherently linked to $Y$. On the other hand, if the protected attribute and the target are not exactly the identical concept universally—*i.e.*, there exists some distribution in which $H(Y|A) > 0$—we can reasonably hope to reduce the bias in a strongly biased dataset by simply collecting some samples from such distributions (denoted the **universal distribution** hereafter; formally defined and analyzed in Sec. 4). Intuitively, when we refer to samples from a universal distribution, we simply mean that it is possible to collect samples that are not extremely biased for a classification task. While collecting the additional data itself is often feasible in practice, collecting target task labels for the additional data can be very time- and cost-intensive because it requires domain expertise. **However, existing methods for reducing attribute bias require both target labels and protected attribute labels to utilize any universal distribution** (Kim et al., 2019; Wang et al., 2020; Nam et al., 2020; Tartaglione et al., 2021; Zhu et al., 2021; Hong & Yang, 2021). We show that it is possible to utilize samples from universal distribution **without any target labels** to reduce bias in a strongly biased dataset.

To that end, we will propose a method that can utilize samples from the universal distribution to filter out protected attributes while maximally preserving all other attributes. We will show in Sec. 6 that our trained method can then be applied to downstream tasks with strongly biased datasets as a simple task-agnostic image preprocessing operation to mitigate the strong bias. We will also show that our method works even when the universal distribution is itself biased (Sec. 6.7 and Fig. 8). The connection between our theoretical findings regarding extreme bias and the proposed method is summarized in Fig. 3. We will discuss the rationale and technical details of the way to *remove* protected attribute and *preserve* all other attributes in the remainder of this section.

Fig. 4 illustrates our proposed method. We assume the inputs are images in explaining our method, which can be readily generalized to other modalities, as shown in the census experiments in Sec. 6. Given an image $x \in \mathcal{X}$ with protected attribute $a \in \mathcal{A}$, we use an encoder $G_{enc} : \mathcal{X} \to \mathcal{Z}$ to map $x$ to latent representation $z = G_{enc}(x)$, and an attribute-conditioned decoder $G_{dec} : \mathcal{Z} \times \mathcal{A} \to \mathcal{X}$ to reconstruct the original image $\hat{x} = G_{dec}(z, a)$ and produce a corresponding filtered image $x' = G_{dec}(z, a')$, where $a' \in \mathcal{A}$ is a constant value for all input images, representing a *neutral value* of the attribute. For example, in Colored MNIST, we choose $a'$ to be a constant RGB color, and in the case of the discrete attribute in CelebA, $a'$ is the uniform categorical distribution. The target of our optimization is $x'$, where we need protected attribute information to be removed (*i.e.*, constant attribute), while all other information about $x$ is preserved.

**Removing Protected Attribute.** To enable generating the filtered image $x'$ by swapping the attribute $a$ with its neutral value $a'$, we need to ensure the representation $z$ and $a$ are disentangled (Bengio et al., 2013; Locatello et al., 2020; Zhu et al., 2022). It is noteworthy that given that the filter is trained on a universal distribution where $H(Y|A) > 0$, there is room for $I(Y; A)$ to decrease while $I(Z; Y)$ is maintained at the same level, according to Theorem 1. This allows for $I(Z; A)$ to be minimized more effectively, resulting in improved bias mitigation. Thus, to achieve disentanglement, we minimize the mutual information loss between their corresponding random variables $Z$ and $A$ where we adopt mutual information neural estimator (Belghazi et al., 2018) and use an auxiliary neural network $T : \mathcal{Z} \times \mathcal{A} \to \mathbb{R}$ for estimating $I(Z; A)$ in Eq. (2):

---

[5]See Appendix 3 for the feasibility of collecting protected attribute labels.

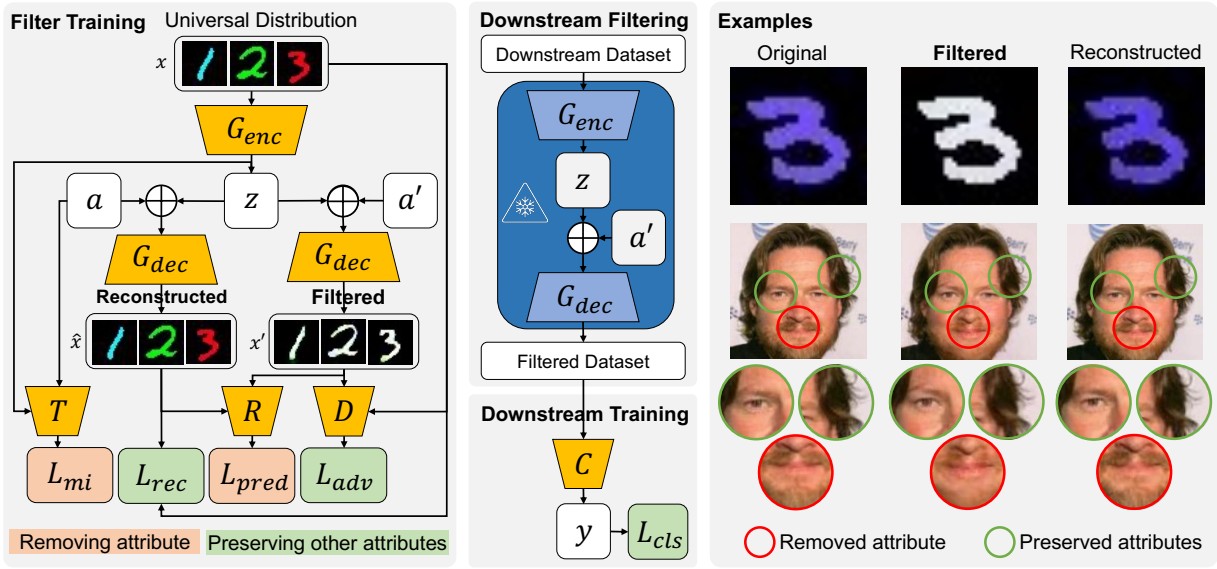

Figure 4: Summary of our method. (Left) Shows the training mechanism of the filter ($G_{dec} \circ G_{enc}$) with samples from a universal distribution, where the protected attribute is removed and other attributes are preserved. (Middle) Shows the use of the filter in a downstream task, where the frozen pretrained filter is first used to remove the protected attribute from the downstream dataset (top), and then the resulting filtered dataset is used to train a classifier $C$ with cross-entropy loss $L_{cls}$ (bottom). (Right) Application of the proposed method to Colored MNIST (*color* as protected attribute) and CelebA (*sex* as protected attribute). The comprehensive empirical results are shown in

$$\mathcal{L}_{mi}^G = \max_T \mathbb{E}_{Z,A} T(z,a) - \log \mathbb{E}_{Z \otimes A} e^{T(z,a)} \tag{2}$$

where $\mathbb{E}_{Z,A}$ and $\mathbb{E}_{Z \otimes A}$ represent the joint and product distribution of latent features and attributes, respectively. Next, to ensure that the reconstruction $\hat{x}$ and the filtered image $x'$ contain the respective attributes, $a$ and $a'$, we introduce a regressor $R : \mathcal{X} \to \mathcal{A}$ trained to achieve classifier guidance generation (Dhariwal & Nichol, 2021) for the loss in Eq. (3):

$$\mathcal{L}_{pred}^G = \min_R \mathcal{L}_{reg}(R(\hat{x}), a) + \mathcal{L}_{reg}(R(x'), a') \tag{3}$$

where $\mathcal{L}_{reg}$ is an appropriate regression loss ($L_2$ loss for continuous attributes and cross-entropy loss for discrete attributes). The loss ensures the generated images contain the respective attributes.

**Preserving Other Attributes.** To ensure the minimal loss of information and preserve other attributes in the filtered image $x'$, we introduce two reconstruction losses inspired by (He et al., 2019). First, an $L_1$ reconstruction loss is applied on the reconstructed image, which can maximally ensure pixel-level information preservation, as shown in Eq. (4):

$$\mathcal{L}_{rec}^G = \min_G \mathbb{E}_{X,\hat{X}} \|x - \hat{x}\|_1 \tag{4}$$

Second, since $L_1$ reconstruction is too strict on the filtered image $x'$, we introduce an adversarial loss for matching it with the original image $x$. We follow WGAN (Arjovsky et al., 2017) with a critic neural network $D : \mathcal{X} \to \mathbb{R}$ for the loss in Eq. (5):

$$\mathcal{L}_{adv}^G = \max_{\|D\|_L = 1} \mathbb{E}_X D(x) - \mathbb{E}_{X'} D(x') \tag{5}$$

where the Lipschitz constraint $\|D\|_L = 1$ on $D$ is enforced through gradient penalty (Gulrajani et al., 2017).

**Overall.** The overall loss that is minimized over $G : \{G_{enc}, G_{dec}\}$ to train the filter is shown in Eq. (6):

$$\mathcal{L}_{total}^G = \min_{G,R} \max_{T,D} (\mathcal{L}_{adv}^G + \lambda_{mi}\mathcal{L}_{mi}^G + \lambda_{pred}\mathcal{L}_{pred}^G + \lambda_{rec}\mathcal{L}_{rec}^G) \tag{6}$$

where $\lambda_{mi}$, $\lambda_{pred}$ and $\lambda_{rec}$ are hyper-parameters that balance losses to optimize $G$. In practice, we alternate between optimizing $T, R, D$ under Eqs. (2), (3) and (5), respectively, and optimizing $G$ under Eq. (6) every other step. After training, the filter $G_{dec} \circ G_{enc} : \mathcal{X} \to \mathcal{X}$ is directly applied in various downstream tasks to remove the protected attribute[6]. In all experiments, we use a two-stage training scheme. First, we train the adversarial filter on either the biased training set itself or samples from a universal distribution (when its availability is assumed). Second, we apply the filter to the biased training set and then train the baseline neural network classifier on the filtered samples with cross-entropy loss. **We provide several ablation studies on the hyper-parameters and the training scheme of our method in Sec. 6.7**.

## 6 Experimental Evaluation

We conduct experiments with an extensive list of existing state-of-the-art attribute bias removal methods based on explicit or implicit mutual information minimization (Kim et al., 2019; Wang et al., 2020; Nam et al., 2020; Tartaglione et al., 2021; Zhu et al., 2021; Hong & Yang, 2021) and further compare our method with several generative model-based approaches (Sauer & Geiger, 2021; Goel et al., 2020; Kim et al., 2021; Ramaswamy et al., 2021), on Colored MNIST as well as two real-world datasets: CelebA (Liu et al., 2015) as an image dataset and Adult (Asuncion & Newman, 2007) as a census dataset. We provide a comprehensive table that summarizes the results of our method across every modality. Please see Sec. 6.6 for results on additional datasets including Waterbirds (Sagawa et al., 2019) and CivilComments-WILDS (Borkan et al., 2019; Koh et al., 2021), Appendix 9 for IMDB (Rothe et al., 2015), and Appendix 14 for training details. **In all experiments, we report average results with one standard deviation over multiple trials** (15 trials in Colored MNIST, 5 in CelebA, 25 in Adult, 5 in Waterbirds, 25 in CivilComments-WILDS, and 5 in IMDB).

Table 1: Overview of the performance of our method in the strong bias region across datasets, reported using AUC of accuracy and relative improvement ($\Delta$).

| Dataset | Method | AUC of Accuracy ↑ | $\Delta$ (%) ↑ |
|---|---|---|---|
| Colored MNIST | Baseline | $60.95_{\pm 6.34}$ | 0.00 |
| | Ours | $77.12_{\pm 2.31}$ | 0.27 |
| | Ours (Universal) | $98.02_{\pm 0.64}$ | 0.61 |
| CelebA | Baseline | $24.67_{\pm 0.72}$ | 0.00 |
| | Ours | $28.90_{\pm 0.94}$ | 0.17 |
| | Ours (FFHQ) | $30.29_{\pm 0.68}$ | 0.23 |
| | Ours (Synthetic) | $30.20_{\pm 0.85}$ | 0.22 |
| Adult | Baseline | $46.36_{\pm 1.54}$ | 0.00 |
| | Ours | $50.29_{\pm 0.44}$ | 0.08 |
| | Ours (Universal) | $53.54_{\pm 0.59}$ | 0.15 |

**Colored MNIST Dataset** is an image dataset of handwritten digits, where each digit is assigned a unique RGB color with a certain variance, studied by these methods (Kim et al., 2019; Tartaglione et al., 2021; Zhu et al., 2021; Ragonesi et al., 2021). The training set consists of 50000 images and the testing set consists of 10000 images with uniformly random color assignment. The color is considered the protected attribute $A$

---

[6]Details of our neutral networks are provided in Appendix 14.

and the digit is the target $Y$. The variance of color in the training set determines the strength of the bias $H(Y|A)$. universal distribution is constructed in a synthetic manner by assigning random colors to digits. The results on this dataset are reported in Fig. 1 and explained in Sec. 1.

Table 2: Performance of attribute bias removal methods under **extreme bias in CelebA dataset** (*TrainEx* training set) to predict *blond hair* (Sec. 6.1). $\Delta$ indicates the difference from baseline, and **Bold** highlights best results. For our method, we report inside parentheses the partially-observable universal distribution used in addition to *TrainEx* for training its filter. Without a universal distribution, none of the methods can effectively remove the bias $I(Z;A)$ compared to baseline.

| Method | Test Accuracy | | Mutual Information | |
|---|---|---|---|---|
| | Unbiased ↑ | Bias-conflicting ↑ | $I(Z;A)$ ↓ | $\Delta$ (%) ↑ |
| Random guess | 50.00 | 50.00 | 0.57 | 0.00 |
| Baseline | $66.11_{\pm 0.32}$ | $33.89_{\pm 0.45}$ | $0.57_{\pm 0.01}$ | 0.00 |
| LNL Kim et al. (2019) | $64.81_{\pm 0.17}$ | $29.72_{\pm 0.26}$ | $0.56_{\pm 0.06}$ | 1.75 |
| DI Wang et al. (2020) | $66.83_{\pm 0.44}$ | $33.94_{\pm 0.65}$ | $0.55_{\pm 0.02}$ | 3.51 |
| LfF Nam et al. (2020) | $64.43_{\pm 0.43}$ | $30.45_{\pm 1.63}$ | $0.57_{\pm 0.03}$ | 0.00 |
| EnD Tartaglione et al. (2021) | $66.53_{\pm 0.23}$ | $31.34_{\pm 0.89}$ | $0.57_{\pm 0.05}$ | 0.00 |
| CSAD Zhu et al. (2021) | $63.24_{\pm 2.36}$ | $29.13_{\pm 1.26}$ | $0.55_{\pm 0.04}$ | 3.51 |
| BCL Hong & Yang (2021) | $65.30_{\pm 0.51}$ | $33.44_{\pm 1.31}$ | $0.56_{\pm 0.07}$ | 1.75 |
| Ours | $66.31_{\pm 0.26}$ | $32.22_{\pm 0.43}$ | $0.55_{\pm 0.01}$ | 3.51 |
| Ours (FFHQ) | $\mathbf{71.53_{\pm 0.67}}$ | $47.17_{\pm 0.72}$ | $0.47_{\pm 0.01}$ | 17.54 |
| Ours (Synthetic) | $71.37_{\pm 0.64}$ | $\mathbf{48.06_{\pm 0.82}}$ | $\mathbf{0.45_{\pm 0.01}}$ | $\mathbf{21.05}$ |

Table 3: Performance of attribute bias removal methods under **extreme bias in Adult** to predict *income* (Sec. 6.1).

| Method | Test Accuracy | | Mutual Information | |
|---|---|---|---|---|
| | Unbiased ↑ | Bias-conflicting ↑ | $I(Z;A)$ ↓ | $\Delta$ (%) ↑ |
| Random guess | 50.00 | 50.00 | 0.69 | 0.00 |
| Baseline | $50.59_{\pm 0.54}$ | $1.19_{\pm 0.83}$ | $0.69_{\pm 0.00}$ | 0.00 |
| LNL Kim et al. (2019) | $50.10_{\pm 0.18}$ | $0.43_{\pm 0.46}$ | $0.69_{\pm 0.01}$ | 0.00 |
| DI Wang et al. (2020) | $50.61_{\pm 0.28}$ | $0.65_{\pm 0.64}$ | $0.69_{\pm 0.01}$ | 0.00 |
| LfF Nam et al. (2020) | $50.33_{\pm 0.34}$ | $0.78_{\pm 0.65}$ | $0.69_{\pm 0.01}$ | 0.00 |
| EnD Tartaglione et al. (2021) | $50.59_{\pm 0.75}$ | $1.18_{\pm 0.96}$ | $0.69_{\pm 0.00}$ | 0.00 |
| CSAD Zhu et al. (2021) | $50.76_{\pm 2.22}$ | $1.43_{\pm 2.46}$ | $0.69_{\pm 0.01}$ | 0.00 |
| BCL Hong & Yang (2021) | $50.83_{\pm 1.34}$ | $0.52_{\pm 0.83}$ | $0.69_{\pm 0.00}$ | 0.00 |
| Ours | $50.09_{\pm 0.81}$ | $0.64_{\pm 1.01}$ | $0.69_{\pm 0.01}$ | 0.00 |
| Ours (Universal) | $\mathbf{74.93_{\pm 0.95}}$ | $\mathbf{57.63_{\pm 1.30}}$ | $\mathbf{0.45_{\pm 0.00}}$ | $\mathbf{34.78}$ |

**CelebA Dataset** (Liu et al., 2015) is an image dataset of human faces studied by these methods (Kim et al., 2019; Wang et al., 2020; Nam et al., 2020; Tartaglione et al., 2021; Zhu et al., 2021; Hong & Yang, 2021). Facial attributes are considered the prediction target $Y$ (*e.g.*, blond hair), and sex is the protected attribute $A$. For each target, there is a notion of *biased samples* – images in which $Y$ is positively correlated with $A$, *e.g.*, images of females with blond hair and males without blond hair – and a notion of *bias-conflicting samples* – images in which $Y$ is negatively correlated with $A$, *e.g.*, images of females without blond hair and males with blond hair. The fraction of bias-conflicting images in the training set determines the strength of the bias $H(Y|A)$. For training, we consider the original training set of CelebA denoted *TrainOri* consisted of 162770 images with $H(Y|A) = 0.36$, and an extreme bias version in which the bias-conflicting samples are removed from the original training set denoted *TrainEx* consisted of 89754 images with $H(Y|A) = 0$. Additionally, we construct 16 training sets between TrainOri and TrainEx by maintaining the number of biased samples and varying the fraction of bias-conflicting samples. For testing, we consider two versions of the original testing set: 1) *Unbiased* consists of 720 images in which all pairs of target and protected attribute labels have the same number of samples, and 2) *Bias-conflicting* consists of 360 images in which biased samples are excluded from the *Unbiased* dataset (only bias-conflicting samples remain). We consider two choices for a universal distribution: 1) appending *TrainEx* with the FFHQ dataset (Karras et al., 2019), and 2)

Table 4: Area under the curve (AUC) in the **strong bias region of CelebA dataset** (Sec. 6.2).

| Method | AUC of Test Accuracy | | AUC of Mutual Information | |
|---|---|---|---|---|
| | Unbiased ↑ | Bias-conflicting ↑ | $I(Z; A) \downarrow$ | $\Delta$ (%) ↑ |
| Random guess | 17.50 | 17.50 | 0.15 | 0.00 |
| Baseline | $24.67_{\pm 0.72}$ | $17.18_{\pm 1.62}$ | $0.15_{\pm 0.01}$ | 0.00 |
| LNL Kim et al. (2019) | $26.81_{\pm 0.97}$ | $21.58_{\pm 0.95}$ | $0.12_{\pm 0.03}$ | 20.00 |
| DI Wang et al. (2020) | $27.53_{\pm 0.92}$ | $23.81_{\pm 0.76}$ | $0.12_{\pm 0.01}$ | 20.00 |
| LfF Nam et al. (2020) | $26.79_{\pm 1.16}$ | $23.78_{\pm 1.24}$ | $0.11_{\pm 0.01}$ | 26.67 |
| EnD Tartaglione et al. (2021) | $27.31_{\pm 0.96}$ | $21.42_{\pm 0.88}$ | $0.12_{\pm 0.03}$ | 20.00 |
| CSAD Zhu et al. (2021) | $27.43_{\pm 1.57}$ | $22.06_{\pm 0.97}$ | $0.12_{\pm 0.02}$ | 20.00 |
| BCL Hong & Yang (2021) | $27.82_{\pm 0.66}$ | $23.53_{\pm 1.32}$ | $0.12_{\pm 0.03}$ | 20.00 |
| Ours | $28.90_{\pm 0.94}$ | $24.61_{\pm 0.79}$ | $0.11_{\pm 0.01}$ | 26.67 |
| Ours (FFHQ) | $\mathbf{30.29_{\pm 0.68}}$ | $25.83_{\pm 1.00}$ | $\mathbf{0.10_{\pm 0.01}}$ | $\mathbf{33.33}$ |
| Ours (Synthetic) | $30.20_{\pm 0.85}$ | $\mathbf{26.04_{\pm 1.22}}$ | $\mathbf{0.10_{\pm 0.01}}$ | $\mathbf{33.33}$ |

appending *TrainEx* with a same-sized synthetic dataset where images are randomly generated using (Li & Abd-Almageed, 2023).

**Adult Dataset** (Asuncion & Newman, 2007) is a census dataset of income which is a well-known fairness benchmark. Income is considered the target $Y$ and sex is the protected attribute $A$. To construct training and testing sets, we follow the setup of CelebA explained above, but we further mitigate the effect of data imbalance and the variation in the total number of training samples. For training, we consider the balanced version of the original training set of Adult denoted *TrainOri* consisted of 7076 records with $H(Y|A) = 0.69$, and an extreme bias version in which the bias-conflicting samples are removed from TrainOri and the same number of biased samples are appended denoted *TrainEx* with $H(Y|A) = 0$ consisted of the same total number (7076) of records as TrainOri. Additionally, we construct 11 training sets between TrainOri and TrainEx by varying the fraction of biased samples in TrainEx while maintaining the total size of the training set. For testing, we consider two versions of the original testing set: 1) *Unbiased* consists of 7076 records in which all pairs of target and protected attribute labels have the same number of samples, and 2) *Bias-conflicting* consists of 3538 records in which biased samples are excluded from the *Unbiased* dataset (only bias-conflicting samples remain). We utilize TrainOri training set, excluding target labels, as a universal distribution.

## 6.1 Analysis of the Extreme Bias Point $H(Y|A) = 0$

In this section, we investigate the consequences of applying attribute bias removal methods at the extreme bias point $H(Y|A) = 0$. We study two aspects of each method, its classification performance (measured by accuracy on Unbiased and Bias-conflicting settings) and its ability to remove bias (measured by estimating $I(Z; A)$ using (Belghazi et al., 2018) on the training set). Ideally, a method must achieve on-par or better accuracy than the baseline while learning a representation $Z$ that does not reflect the attribute bias present in the training set, hence successfully removing the bias, *i.e.*, $I(Z; A) = 0$. However, as shown in Tabs. 2 and 3, without a universal distribution, none of the bias removal methods can significantly reduce the bias $I(Z; A)$ in the extreme bias setting in either CelebA or Adult datasets. These observations are explained by Proposition 1 which states that maintaining classification performance above random guess while achieving $I(Z; A) = 0$ at $H(Y|A) = 0$ is impossible. Note that the methods achieve better than random accuracy because they do not completely remove the bias.

When given access to a universal distribution, we observe that our method can significantly improve the performance and the amount of removed bias in both synthetic (Fig. 1) and real-world datasets (Tabs. 2 and 3). Note that none of the existing methods can directly utilize the access to the universal distribution due to the lack of target labels which is required by these methods. Nonetheless, it is possible to enable all methods to utilize the additional distribution by using pseudo-labeling, which we will explore in Sec. 6.3.

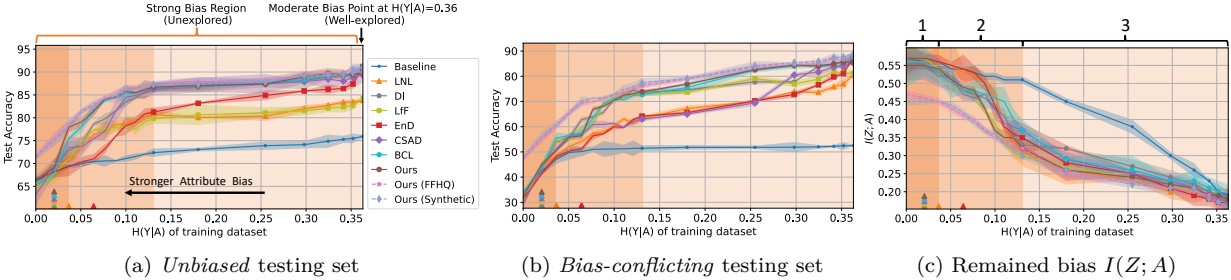

(a) *Unbiased* testing set  (b) *Bias-conflicting* testing set  (c) Remained bias $I(Z; A)$

Figure 5: Accuracy and mutual information under **different bias strengths in CelebA** (Sec. 6.2). As the attribute bias in the training dataset becomes stronger (right to left on the x-axis), the performance of all methods degrades. All methods, except ours with universal distribution, eventually become the same as the baseline classifier (at the breaking point labeled by ▲).

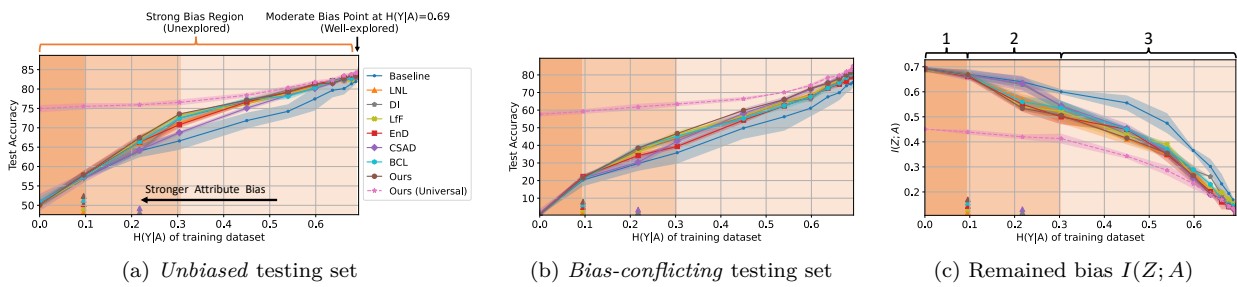

(a) *Unbiased* testing set  (b) *Bias-conflicting* testing set  (c) Remained bias $I(Z; A)$

Figure 6: Accuracy and mutual information under **different bias strengths in Adult dataset** (Sec. 6.2).

## 6.2  Analysis of the Strong Bias Region $H(Y|A) > 0$

In this section, we go beyond the extreme bias point, and more generally investigate the consequences of applying bias removal methods on the entire range of bias strength, *i.e.*, connecting the extreme bias training setting (TrainEx) we studied in Sec. 6.1 to the moderate bias in the original training setting (TrainOri) commonly studied in existing methods. We again study two aspects of each method, its classification performance (measured by accuracy on Unbiased and Bias-conflicting settings) and its ability to remove bias (measured by estimating $I(Z; A)$ using (Belghazi et al., 2018) on the training set).

Without access to a universal distribution, in Figs. 5 and 6, we observe a performance decline across all methods as bias strength increases, in both CelebA and Adult datasets, similar to our prior observation in Colored MNIST in Fig. 1. This observation aligns with Theorem 1, which states that bias strength determines an upper bound on the best performance of bias removal methods regardless of dataset and method. Furthermore, in Figs. 5c and 6c, we use breaking points (as defined in Sec. 1) to approximately divide the strong bias region into three phases and explain the observed changes in the performance of methods from the perspective of Theorem 1. In phase 1, as $H(Y|A)$ increases from zero to the breaking point (bias strength decreases), we observe that the remained attribute bias $I(Z; A)$ is not minimized because of the trade-off between best attainable performance $I(Z; Y)$ and attribute bias removal when bias is very strong: the methods choose to increase accuracy towards the best attainable accuracy $I(Z; Y)$ rather than removing attribute bias (this choice is most likely due to the larger weight on the accuracy term in their objectives). Then, in phase 2, as $H(Y|A)$ increases through the breaking point (bias strength decreases further), the methods start to minimize the remained attribute bias $I(Z; A)$ because the upper bound on best attainable performance $I(Z; Y)$ is now large enough to avoid the trade-off between accuracy and attribute bias removal. Finally, in phase 3, as $H(Y|A)$ further departs from the breaking point, accuracy gradually approaches its best attainable performance, while remained attribute bias $I(Z; A)$ is minimized further below that of the baseline because the weaker bias strength now allows the model to distinguish $Y$ from $A$ so that minimizing attribute bias and maximizing accuracy do not compete.

Table 5: Area under the curve (AUC) in the **strong bias region of Adult dataset** (Sec. 6.2).

| Method | AUC of Test Accuracy | | AUC of Mutual Information | |
|---|---|---|---|---|
| | Unbiased ↑ | Bias-conflicting ↑ | $I(Z;A) \downarrow$ | $\Delta$ (%) ↑ |
| Random guess | 34.00 | 34.00 | 0.38 | 0.00 |
| Baseline | $46.36_{\pm 1.54}$ | $27.38_{\pm 4.64}$ | $0.38_{\pm 0.02}$ | 0.00 |
| LNL Kim et al. (2019) | $48.36_{\pm 0.49}$ | $31.41_{\pm 1.25}$ | $0.32_{\pm 0.02}$ | 15.79 |
| DI Wang et al. (2020) | $48.38_{\pm 0.53}$ | $31.30_{\pm 1.09}$ | $0.34_{\pm 0.01}$ | 10.53 |
| LfF Nam et al. (2020) | $48.57_{\pm 0.50}$ | $31.64_{\pm 1.17}$ | $0.33_{\pm 0.02}$ | 13.16 |
| EnD Tartaglione et al. (2021) | $48.42_{\pm 0.71}$ | $30.10_{\pm 1.08}$ | $0.32_{\pm 0.02}$ | 15.79 |
| CSAD Zhu et al. (2021) | $47.58_{\pm 0.69}$ | $31.11_{\pm 1.54}$ | $0.34_{\pm 0.02}$ | 10.53 |
| BCL Hong & Yang (2021) | $48.54_{\pm 0.73}$ | $31.20_{\pm 1.17}$ | $0.33_{\pm 0.01}$ | 13.16 |
| Ours | $50.29_{\pm 0.44}$ | $33.63_{\pm 0.99}$ | $0.31_{\pm 0.01}$ | 18.42 |
| Ours (Universal) | $\mathbf{53.54_{\pm 0.59}}$ | $\mathbf{45.16_{\pm 1.18}}$ | $\mathbf{0.25_{\pm 0.01}}$ | $\mathbf{34.21}$ |

Table 6: Effect of **pseudo-labeling** on attribute bias removal methods under **extreme bias in CelebA** (Sec. 6.3). The baseline trained on the extreme bias dataset (*TrainEx*) is listed for reference. All other methods are trained on the combination of *TrainEx* and FFHQ pseudo-labeled by a classifier pretrained on *TrainEx*. With pseudo-labeling, all methods outperform the baseline, with our proposed method achieving the best.

| Method | Test Accuracy | | Mutual Information | |
|---|---|---|---|---|
| | Unbiased ↑ | Bias-conflicting ↑ | $I(Z;A) \downarrow$ | $\Delta$ (%) ↑ |
| Baseline (TrainEx) | $66.11_{\pm 0.32}$ | $33.89_{\pm 0.45}$ | $0.57_{\pm 0.01}$ | 0.00 |
| Baseline | $67.02_{\pm 0.78}$ | $35.25_{\pm 1.32}$ | $0.48_{\pm 0.01}$ | 15.79 |
| LNL Kim et al. (2019) | $67.47_{\pm 0.34}$ | $40.56_{\pm 1.24}$ | $0.43_{\pm 0.04}$ | 24.56 |
| DI Wang et al. (2020) | $70.61_{\pm 0.58}$ | $46.89_{\pm 0.83}$ | $0.39_{\pm 0.03}$ | 31.58 |
| LfF Nam et al. (2020) | $69.42_{\pm 0.61}$ | $45.54_{\pm 1.26}$ | $0.41_{\pm 0.04}$ | 28.07 |
| EnD Tartaglione et al. (2021) | $67.65_{\pm 0.34}$ | $42.85_{\pm 0.65}$ | $0.42_{\pm 0.01}$ | 26.32 |
| CSAD Zhu et al. (2021) | $68.18_{\pm 0.16}$ | $46.51_{\pm 0.81}$ | $0.39_{\pm 0.02}$ | 31.58 |
| BCL Hong & Yang (2021) | $70.43_{\pm 0.71}$ | $46.86_{\pm 1.61}$ | $0.39_{\pm 0.03}$ | 31.58 |
| Ours | $\mathbf{72.05_{\pm 0.86}}$ | $\mathbf{48.72_{\pm 0.56}}$ | $\mathbf{0.38_{\pm 0.01}}$ | $\mathbf{33.33}$ |

To better quantify the performance and compare different methods across the entire strong bias region, in Tabs. 4 and 5, we report the area under the curves in Figs. 5 and 6, respectively. We observe that our proposed method achieves the best performance in both datasets and in all metrics (accuracy and bias removal). In addition, it achieves better or on-par breaking points with existing methods. The same observation holds in the Colored MNIST dataset in Fig. 1. This shows that even though we designed our method to be able to utilize a universal distribution, it can outperform existing methods even without access to such a dataset as well, suggesting that it can be used as a state-of-the-art bias removal method in all settings. We conjecture that this advantage is because we explicitly encourage the filter to maximally preserve information, whereas in other methods the mutual information minimization can remove any information that is not used by the jointly trained classifier, potentially removing too much information early in training when the classifier is relying on only a few features, thus trapping it in local minima.

With access to a universal distribution, we observe that our method can now significantly improve the performance and the amount of removed bias in both synthetic (Colored MNIST in Fig. 1) and real-world datasets (CelebA and Adult in Tabs. 4 and 5). Note that none of the existing methods can directly utilize the universal distribution due to the lack of target labels which is required by these methods. This shows that our method can effectively utilize a partially-observable universal distribution to improve attribute bias removal.

## 6.3 Pseudo-Labeling of the Universal Distribution

While existing methods cannot directly utilize a partially-observable universal distribution with missing target labels because they require both the protected attribute and the target labels to compute their objectives, it is possible to convert the partially-observable universal distribution to an approximately fully-

Table 7: Effect of **pseudo-labeling** on attribute bias removal methods under **extreme bias in Adult** (Sec. 6.3).

| Method | Test Accuracy | | Mutual Information | |
|---|---|---|---|---|
| | Unbiased ↑ | Bias-conflicting ↑ | $I(Z;A) \downarrow$ | $\Delta$ (%) ↑ |
| Baseline (TrainEx) | $50.59_{\pm 0.54}$ | $1.19_{\pm 0.83}$ | $0.69_{\pm 0.00}$ | 0.00 |
| Baseline | $60.86_{\pm 0.13}$ | $22.21_{\pm 0.42}$ | $0.54_{\pm 0.04}$ | 21.74 |
| LNL Kim et al. (2019) | $68.46_{\pm 0.43}$ | $46.75_{\pm 0.41}$ | $0.46_{\pm 0.03}$ | 33.33 |
| DI Wang et al. (2020) | $73.25_{\pm 0.32}$ | $54.14_{\pm 0.62}$ | $0.42_{\pm 0.02}$ | 39.13 |
| LfF Nam et al. (2020) | $70.86_{\pm 0.72}$ | $51.25_{\pm 0.56}$ | $0.44_{\pm 0.02}$ | 36.23 |
| EnD Tartaglione et al. (2021) | $73.78_{\pm 1.21}$ | $56.75_{\pm 1.13}$ | $0.43_{\pm 0.03}$ | 37.68 |
| CSAD Zhu et al. (2021) | $72.93_{\pm 1.62}$ | $56.82_{\pm 1.95}$ | $0.42_{\pm 0.03}$ | 39.13 |
| BCL Hong & Yang (2021) | $73.75_{\pm 0.63}$ | $57.52_{\pm 1.43}$ | $0.41_{\pm 0.02}$ | 40.58 |
| Ours | $\mathbf{76.35_{\pm 0.31}}$ | $\mathbf{60.56_{\pm 1.82}}$ | $\mathbf{0.39_{\pm 0.00}}$ | **43.48** |

Table 8: Accuracy of attribute bias removal methods under extreme bias and moderate bias in **all 23 non-sex-related downstream tasks of CelebA dataset** (Sec. 6.4). Our proposed method achieves the best performance, both with and without access to the universal distribution, showing that its trained filter has preserved the information of the other 23 attributes while removing the protected attribute (*i.e.*, sex in CelebA). See Appendix 7 for separate per-task results.

| Method | Extreme Bias Training (*TrainEx*) | | Moderate Bias Training (*TrainOri*) | |
|---|---|---|---|---|
| | Unbiased ↑ | Bias-conflicting ↑ | Unbiased ↑ | Bias-conflicting ↑ |
| Baseline | $59.03_{\pm 0.96}$ | $21.53_{\pm 1.42}$ | $78.08_{\pm 0.82}$ | $71.85_{\pm 1.04}$ |
| LNL Kim et al. (2019) | $55.84_{\pm 0.31}$ | $18.81_{\pm 0.53}$ | $78.43_{\pm 0.75}$ | $75.03_{\pm 1.27}$ |
| DI Wang et al. (2020) | $59.73_{\pm 0.43}$ | $22.03_{\pm 0.42}$ | $80.83_{\pm 0.54}$ | $76.45_{\pm 0.42}$ |
| LfF Nam et al. (2020) | $56.12_{\pm 0.35}$ | $20.45_{\pm 1.54}$ | $79.31_{\pm 0.68}$ | $75.82_{\pm 1.73}$ |
| EnD Tartaglione et al. (2021) | $58.32_{\pm 0.47}$ | $20.48_{\pm 0.89}$ | $81.14_{\pm 1.61}$ | $77.03_{\pm 2.73}$ |
| CSAD Zhu et al. (2021) | $54.65_{\pm 1.43}$ | $18.93_{\pm 2.07}$ | $80.45_{\pm 1.82}$ | $76.20_{\pm 2.94}$ |
| BCL Hong & Yang (2021) | $59.28_{\pm 0.58}$ | $22.16_{\pm 0.53}$ | $81.02_{\pm 0.12}$ | $77.81_{\pm 1.83}$ |
| Ours | $60.13_{\pm 0.27}$ | $22.45_{\pm 1.52}$ | $81.62_{\pm 1.46}$ | $78.76_{\pm 2.84}$ |
| Ours (FFHQ) | $63.43_{\pm 0.98}$ | $34.98_{\pm 1.93}$ | $82.62_{\pm 1.12}$ | $79.78_{\pm 1.54}$ |
| Ours (Synthetic) | $\mathbf{63.76_{\pm 1.03}}$ | $\mathbf{36.29_{\pm 1.24}}$ | $\mathbf{83.24_{\pm 1.03}}$ | $\mathbf{80.23_{\pm 1.84}}$ |

observable distribution using pseudo labels: labels collected using a pretrained classifier. This enables all methods to utilize the additional data available in universal distribution. To investigate the effectiveness of pseudo-labeling, we first train a baseline classifier on the observed biased dataset TrainEx – ResNet18 (He et al., 2016) for CelebA and a three-layer MLP for Adult – then we use this trained classifier to label samples of the universal distribution, and finally provide all methods with the original biased dataset extended with the pseudo-labeled samples of the universal distribution. The results are reported in Tabs. 6 and 7. We observe that pseudo-labeling improves the performance of all methods (compared to Tabs. 2 and 3), and that our method still achieves the best performance in both datasets, showing that our proposed method can be used together with pseudo-labeling to provide additional gains. We attribute this to the target-agnostic design of our method, which diminishes the reliance on the quality of pseudo-labels.

## 6.4 Application to Various Downstream Tasks

In this section, we investigate whether our trained filter can be applied to various downstream target prediction tasks, *i.e.*, whether it can in fact maximally preserve information while removing the attribute bias. To this end, in Tab. 8, we report the average performance of our method on all 23 non-sex-related downstream tasks in CelebA, in both the extreme and moderate attribute bias settings (sex is considered the protected attribute). Note that the filtering mechanism in the proposed method is only trained once, and then reused in all downstream tasks without retraining. We observe that our proposed method achieves the best performance, even without access to the universal distribution. The results for individual tasks are reported in Appendix 7. This observation suggests that our proposed method can maintain information regarding all other attributes when removing the protected attribute.

Table 9: Accuracy of **generative model-based methods** under extreme bias and moderate bias in CelebA dataset to predict *blond hair* (Sec. 6.5). For our method, we report inside parentheses the partially-observable universal distribution used in addition to *TrainEx* for training its filter. Our method performs better than generative model-based methods, while it uses only half the size of the classifier training sets that generative model-based methods require.

| Method | Extreme Bias Training (*TrainEx*) | | | Moderate Bias Training (*TrainOri*) | | |
|---|---|---|---|---|---|---|
| | Size of Classifier Training Set ↓ | Unbiased ↑ | Bias-conflicting ↑ | Size of Classifier Training Set ↓ | Unbiased ↑ | Bias-conflicting ↑ |
| Baseline | 89754 | $66.11_{\pm0.32}$ | $33.89_{\pm0.45}$ | 162770 | $75.92_{\pm0.35}$ | $52.52_{\pm0.19}$ |
| CGN Sauer & Geiger (2021) | 89754×2 | $63.38_{\pm1.34}$ | $31.46_{\pm1.42}$ | 162770×2 | $82.65_{\pm1.82}$ | $79.81_{\pm1.80}$ |
| CAMEL Goel et al. (2020) | 89754×2 | $64.23_{\pm1.82}$ | $32.81_{\pm1.18}$ | 162770×2 | $86.45_{\pm1.17}$ | $82.67_{\pm1.47}$ |
| BiaSwap Kim et al. (2021) | 89754×2 | $65.97_{\pm1.12}$ | $33.67_{\pm1.65}$ | 162770×2 | $88.83_{\pm1.61}$ | $85.45_{\pm1.42}$ |
| GAN-Debiasing Ramaswamy et al. (2021) | 89754×2 | $66.83_{\pm1.73}$ | $32.18_{\pm1.38}$ | 162770×2 | $88.34_{\pm2.05}$ | $85.27_{\pm1.13}$ |
| Ours | 89754 | $66.31_{\pm0.26}$ | $32.22_{\pm0.43}$ | 162770 | $89.81_{\pm0.45}$ | $85.29_{\pm1.54}$ |
| Ours (FFHQ) | 89754 | $\mathbf{71.53_{\pm0.67}}$ | $47.17_{\pm0.72}$ | 162770 | $\mathbf{90.86_{\pm0.87}}$ | $88.06_{\pm0.91}$ |
| Ours (Synthetic) | 89754 | $71.37_{\pm0.64}$ | $\mathbf{48.06_{\pm0.82}}$ | 162770 | $90.01_{\pm0.65}$ | $\mathbf{88.72_{\pm1.16}}$ |

## 6.5 Comparison with Generative Dataset Augmentation

To remove attribute bias, an alternative to our method of filtering the samples in a biased dataset, is to augment the dataset with attribute-flipped samples. Here, we investigate how our method performs compared to state-of-the-art generative model-based methods for attribute flipping (Sauer & Geiger, 2021; Goel et al., 2020; Kim et al., 2021; Ramaswamy et al., 2021). These methods differ from our method in two main aspects: 1) similar to MI-based methods, they require both target and attribute labels to apply attribute flipping, making them incompatible with a partially-observable universal distribution where target labels are missing; 2) they mitigate bias by augmenting the dataset with attribute-flipped samples (rather than filtering the samples), which requires more augmented samples depending on the number of protected attribute values. For example, in CelebA dataset, protected attribute is binary (sex) so they need to increase the dataset size by a factor of two, whereas in Colored MNIST, protected attribute can take ten RGB colors so they need to increase the dataset size by 10 times. In Tab. 9, we report the performance of generative model-based methods. In moderate bias setting, our method achieves better average accuracy than generative model-based methods, with and without using universal distribution. In the extreme bias setting, without access to a universal distribution, none of the methods can outperform the baseline, consistent with our prior observations in Tabs. 2 and 3. Given access to a universal distribution, our method achieves the best average accuracy. These observations provide further evidence that our method is the most effective overall solution for mitigating attribute bias of various strengths, both with and without access to samples from a universal distribution.

## 6.6 Application to Various Protected Attributes and Modalities

We investigate the applicability of our method across various protected attributes and modalities. In addition to Colored MNIST, CelebA, and Adult, which we analyzed in previous sections, we include two additional benchmark datasets to assess the effectiveness of our method in attribute bias removal: Waterbirds (Sagawa et al., 2019) and CivilComment-WILDS (Borkan et al., 2019). We compare our method with the attribute bias removal methods that have specifically studied these two datasets (Nam et al., 2020; Sagawa et al., 2019; Creager et al., 2021; Liu et al., 2021; Zhang et al., 2022). **In these datasets, the training and testing sets are similarly biased,[7] therefore the common criterion for the effectiveness of a bias removal method is to have similar average accuracy with the baseline classifier while having much higher worst-group accuracy** – the group where the baseline performs the worst. A summary of all datasets considered in this work, their respective modalities, and the evaluated protected attributes are provided in Tab. 10.

**Waterbirds** (Sagawa et al., 2019) is an image dataset of various bird species, where the classification target is either waterbird or landbird and the protected attribute is either water background or land background. Attribute bias arises since the training set contains more instances of waterbirds with water backgrounds and

---

[7]We follow the setup of (Sagawa et al., 2019), in which a weighted average accuracy is computed in the testing set, where the weights reflect the size of the groups in the training set, hence the same bias in sample frequency.

Table 10: Summary of all datasets used to evaluate our method across various protected attributes and modalities.

| Name | Modality | Protected Attribute | Prediction Target |
|---|---|---|---|
| Colored MNIST Kim et al. (2019) | Image | Color | Digit |
| CelebA Liu et al. (2015) | Image | Sex | Facial attributes |
| Adult Dua & Graff (2017) | Tabular | Sex | Income |
| Waterbirds Sagawa et al. (2019) | Image | Background | Waterbirds or landbirds |
| CivilComment-WILDS Koh et al. (2021) | Text | Demographic identities | Toxic or non-toxic |

Table 11: Average and worst-group test accuracies in **Waterbirds and CivilComments-WILDS** (Sec. 6.6).

| Model | Waterbirds | | CivilComments-WILDS | |
|---|---|---|---|---|
| | Average | Worst-group | Average | Worst-group |
| Baseline | $97.26_{\pm0.97}$ | $62.60_{\pm0.27}$ | $92.14_{\pm0.38}$ | $58.63_{\pm1.73}$ |
| LfF Nam et al. (2020) | $91.22_{\pm0.85}$ | $78.04_{\pm1.83}$ | $92.52_{\pm0.91}$ | $58.81_{\pm1.23}$ |
| Group DRO Sagawa et al. (2019) | $92.02_{\pm0.62}$ | $89.92_{\pm0.63}$ | $88.91_{\pm0.28}$ | $69.84_{\pm2.39}$ |
| EIIL Creager et al. (2021) | $96.52_{\pm0.21}$ | $77.19_{\pm1.03}$ | $90.48_{\pm0.23}$ | $67.01_{\pm2.42}$ |
| JTT Liu et al. (2021) | $89.34_{\pm0.66}$ | $83.82_{\pm1.23}$ | $91.14_{\pm0.34}$ | $69.26_{\pm0.89}$ |
| CNC Zhang et al. (2022) | $88.51_{\pm0.34}$ | $90.93_{\pm0.11}$ | $81.74_{\pm0.52}$ | $68.92_{\pm2.09}$ |
| Ours | $93.37_{\pm0.81}$ | $91.06_{\pm1.58}$ | $91.26_{\pm0.95}$ | $69.51_{\pm0.71}$ |
| Ours (Universal) | $94.24_{\pm0.92}$ | $\mathbf{93.21_{\pm1.43}}$ | $92.42_{\pm1.43}$ | $\mathbf{70.25_{\pm0.56}}$ |

Table 12: Removing protected attribute analysis (Sec. 6.7): Accuracy of **protected attribute prediction (lower is better)** on the *Unbiased* testing set for sex classification in CelebA. Our filter is trained on the original training set. The vanilla baseline performance is $98.25_{\pm0.13}$. Bold shows the fixed hyper-parameters while others vary.

| $\lambda_{mi}$ | 0 | 10 | 25 | **50** | 60 |
|---|---|---|---|---|---|
| Ours | $95.36_{\pm0.43}$ | $90.78_{\pm0.74}$ | $86.42_{\pm0.54}$ | $84.74_{\pm0.38}$ | $84.02_{\pm0.23}$ |

| $\lambda_{pred}$ | 0 | 10 | 25 | **50** | 60 |
|---|---|---|---|---|---|
| Ours | $97.27_{\pm0.36}$ | $91.13_{\pm0.54}$ | $87.81_{\pm0.87}$ | $84.74_{\pm0.38}$ | $83.45_{\pm0.41}$ |

| $\lambda_{rec}$ | 0 | 10 | 50 | **100** | 110 |
|---|---|---|---|---|---|
| Ours | $70.89_{\pm0.27}$ | $76.21_{\pm0.83}$ | $81.48_{\pm0.61}$ | $84.74_{\pm0.38}$ | $85.09_{\pm0.86}$ |

Table 13: Preserving other attributes analysis (Sec. 6.7): Accuracy of **target prediction (higher is better)** on the *Unbiased* testing set of all 23 non-sex-related downstream tasks of CelebA. Our filter is trained on the original training set. The vanilla baseline performance is $78.08_{\pm0.82}$. Bold shows the fixed hyper-parameters while others vary.

| $\lambda_{mi}$ | 0 | 10 | 25 | **50** | 60 |
|---|---|---|---|---|---|
| Ours | $76.91_{\pm0.43}$ | $78.21_{\pm0.81}$ | $79.98_{\pm1.21}$ | $81.62_{\pm1.46}$ | $80.72_{\pm0.71}$ |

| $\lambda_{pred}$ | 0 | 10 | 25 | **50** | 60 |
|---|---|---|---|---|---|
| Ours | $73.54_{\pm0.17}$ | $76.83_{\pm0.55}$ | $78.39_{\pm0.49}$ | $81.62_{\pm1.46}$ | $79.82_{\pm0.62}$ |

| $\lambda_{rec}$ | 0 | 10 | 50 | **100** | 110 |
|---|---|---|---|---|---|
| Ours | $43.83_{\pm0.46}$ | $60.81_{\pm0.51}$ | $71.43_{\pm0.83}$ | $81.62_{\pm1.46}$ | $81.48_{\pm0.23}$ |

landbirds with land backgrounds compared to other combinations. To construct samples from a universal distribution, we ensure an even number of landbirds and waterbirds on both land and water backgrounds by utilizing provided pixel-level segmentation masks to extract each bird from its original background and then placing it onto water background or land background sourced from the Places dataset (Zhou et al., 2017). We observe in Tab. 11 that the baseline, which focuses on minimizing the average training loss without applying any debiasing techniques, achieves the best weighted average accuracy but results in a significantly poor worst-group accuracy. This is because waterbirds with land backgrounds (*i.e.*, the worst group) are rare in the training set, while waterbirds with water backgrounds are sufficiently represented. As a result, the baseline is biased towards using the background for bird species prediction, which drastically sacrifices the performance for the minority group (*e.g.*, waterbirds with land background) to achieve a good performance in the majority group (*e.g.*, waterbirds with water background), thereby achieving a better weighted average performance. In contrast, our method when trained on the same dataset as other methods, achieves the best worst-group performance (91.06%) with a small drop in average accuracy (3.89%) compared to the baseline. With access to the universal distribution, our method's worst-group performance is further improved to 93.21%, and its drop in average accuracy is also further reduced to 3.02%.

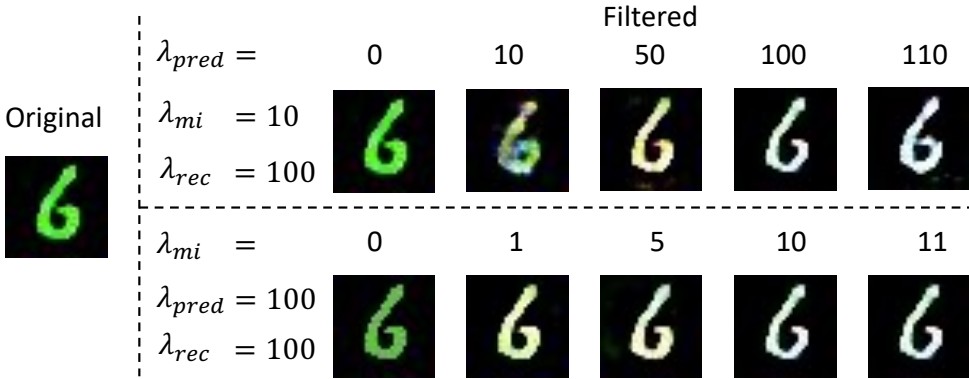

Figure 7: Visual effect of our **hyper-parameters** in removing the protected attribute (color) in Colored MNIST (Sec. 6.7).

**CivilComment-WILDS** (Borkan et al., 2019) is a text dataset consisting of online comments. This text dataset is aimed at classifying online comments as toxic or non-toxic, with target labels often spuriously correlated with mentions of certain demographic identities. To construct samples from a universal distribution, we evenly sample from all 16 groups in this dataset, excluding target labels. We observe in Tab. 11 that, in the absence of samples from the universal distribution, our method performs on-par with other methods; when such samples are available, our method achieves the best worst-group accuracy while having better or on-par average accuracy compared to others. Additionally, in Tab. 11, we observe a significant gap in worst-group accuracy for all methods when comparing CivilComments-WILDS to Waterbirds. We hypothesize that this occurs because, in Waterbirds, each image has a unique background label, whereas, in CivilComments-WILDS, multiple demographic identities may be mentioned in a single comment, making bias mitigation more challenging.

### 6.7 Ablation Studies

**Removing Protected Attribute.** Our method achieves this using the mutual information loss ($\mathcal{L}_{mi}$) and attribute prediction loss ($\mathcal{L}_{pred}$), with weight coefficients $\lambda_{mi}$ and $\lambda_{pred}$, respectively. To qualitatively study the importance of each loss, in Fig. 7, we train our filter on the Colored MNIST dataset with varying coefficients, and observe that if either coefficient is zero, the color is not successfully removed from the digit, thus both $\mathcal{L}_{mi}$ and $\mathcal{L}_{pred}$ are necessary to eliminate the information of protected attributes. Furthermore, to quantitatively measure the importance of each loss in removing the protected attribute, we first train our filter on the CelebA original training set (TrainOri) with varying coefficients, then use it to filter the dataset, and finally measure the attribute prediction accuracy of the baseline classifier trained on the filtered dataset to predict the protected attribute: the lower the attribute prediction accuracy, the better the attribute bias removal. In Tab. 12, we observe that increasing the coefficients of these two losses reduces the attribute prediction accuracy, thus improving attribute bias removal. Additionally, we observe that increasing the coefficient of the reconstruction loss ($\mathcal{L}_{rec}$) results in weaker attribute bias removal (higher attribute prediction accuracy). The recommended coefficients used in all experiments are displayed in bold.

**Preserving Other Attributes.** Our method achieves this using the reconstruction loss ($\mathcal{L}_{rec}$) with weight coefficient $\lambda_{mi}$, and the adversarial loss ($\mathcal{L}_{pred}$) with a constant weight coefficient of 1. To quantitatively measure the importance of the reconstruction loss in preserving other attributes, we first train our filter on the CelebA original training set (TrainOri) with varying coefficients, then use it to filter the dataset, and finally measure the average target prediction accuracy of 23 classifiers for each non-sex-related attribute trained on the filtered dataset to predict the 23 non-sex-related targets in CelebA: the higher the target prediction accuracy, the better preserved the other attributes when removing sex. In Tab. 12, we observe that with a proper choice of $\lambda_{rec}$ their performance on filtered images is consistent with original images, which indicates all relevant facial attributes are preserved. Additionally, we observe that increasing the

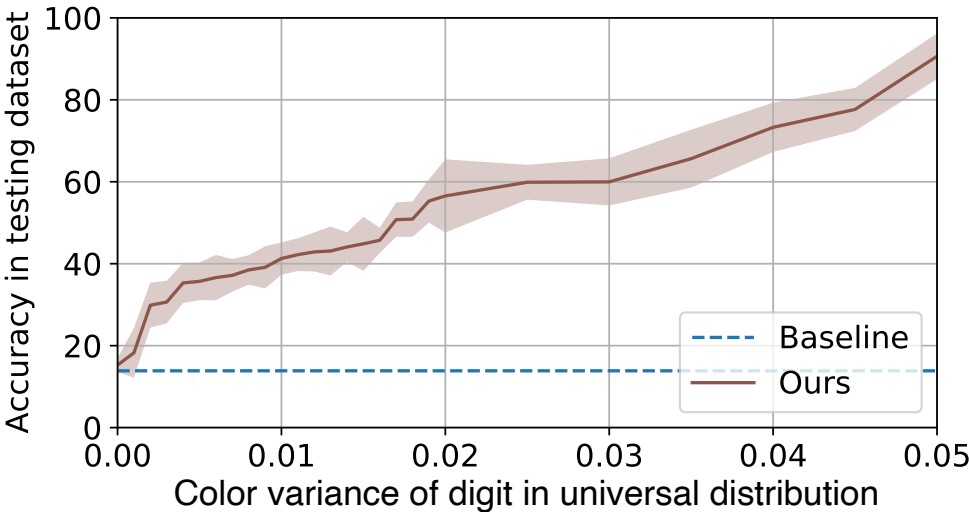

Figure 8: The effect of **bias strength in the universal distribution** on our method in the extreme bias setting (corresponding to the zero location on the horizontal axis in Fig. 1). The baseline classifier and other bias removal methods have constant accuracy (*dashed line*) because they cannot use the partially-observable universal distribution (lacks target labels).

coefficients of the bias removal losses $\lambda_{mi}, \lambda_{pred}$ improves the target prediction accuracy; we hypothesize that this is because the classifier trained on a biased dataset might employ the protected attribute (*e.g.*, sex) as a proxy during training, leading to lower accuracy on datasets without such correlation (Unbiased), and therefore, upon successful removal of sex-related information, an improvement in non-sex-related attribute classification accuracy is observed in Tab. 13. The recommended coefficients used in all experiments are displayed in bold.

**Bias in Universal Distribution.** We aim to investigate the sensitivity of our filter training to the amount of attribute bias in the universal distribution itself, namely $H_q(Y|A)$. To that end, we consider the extreme bias setting in Colored MNIST dataset – where no existing method can outperform the baseline except for our method when using the universal distribution – and measure how the performance of our method varies when we gradually increase the strength of attribute bias in the universal distribution (*i.e.*, decrease color variance). In Fig. 8, we observe that our method can outperform the baseline as long as the bias in the universal distribution ($H_q(Y|A)$) is moderately larger than zero. Consistent with this observation, we also observed in CelebA experiments that using an outside image dataset (FFHQ) as samples from a universal distribution is effective in boosting performance even though we have not explicitly made the dataset unbiased.

**Two-Stage Training and End-to-End Training.** In all experiments detailed in the previous sections, we use a two-stage training scheme for our method. Initially, the filter is trained using samples from the universal distributions and then applied to the classifier training set to obtain filtered samples. Subsequently, the baseline classifier is trained on these filtered samples. In this section, we examine the performance of the proposed method under end-to-end training and compare it with the two-stage training scheme (shown in Fig. 4). In Tab. 14, we observe that the two-stage training scheme outperforms the end-to-end training scheme on average, particularly under stronger attribute bias. We conjecture that this is because, in an end-to-end training scheme, the filter parameters are also updated to minimize the classification loss at the output of the classifier. When the training data is highly biased, this additional update can amplify the bias in the filter output itself, thereby compromising its role in removing (normalizing) the protected attribute. Furthermore, the two-stage training allows the filter to be trained without target labels to further boost its performance (see the rows for *Filter Training Set is Universal* in Tab. 14).

Table 14: Accuracy of target prediction in all 23 non-sex-related downstream tasks of CelebA dataset, **with and without the two-stage training scheme** (Sec. 6.7).

| Method | Filter Training Set | Classifier Training Set | Unbiased ↑ | Bias-conflicting ↑ |
|---|---|---|---|---|
| Baseline | - | Extreme Bias | $59.03_{\pm 0.96}$ | $21.53_{\pm 1.42}$ |
| Ours (end-to-end) | - | Extreme Bias | $59.15_{\pm 1.04}$ | $21.82_{\pm 1.73}$ |
| Ours (two-stage) | Extreme Bias | Extreme Bias | $60.13_{\pm 0.27}$ | $22.45_{\pm 1.52}$ |
| Ours (two-stage) | Universal | Extreme Bias | $\mathbf{63.76_{\pm 1.03}}$ | $\mathbf{36.29_{\pm 1.24}}$ |
| Baseline | - | Moderate Bias | $78.08_{\pm 0.82}$ | $71.85_{\pm 1.04}$ |
| Ours (end-to-end) | - | Moderate Bias | $81.02_{\pm 0.66}$ | $77.91_{\pm 1.33}$ |
| Ours (two-stage) | Moderate Bias | Moderate Bias | $81.62_{\pm 1.46}$ | $78.76_{\pm 2.84}$ |
| Ours (two-stage) | Universal | Moderate Bias | $\mathbf{83.24_{\pm 1.03}}$ | $\mathbf{80.23_{\pm 1.84}}$ |

# 7 Approach to Utilize Non-Observable Universal Distribution

In the previous sections, we show that in extreme bias $H(Y|A) = 0$, the existence of an universal distribution is necessary to overcome the trade-off between attribute bias removal $I(Z; A)$ and the best attainable performance $I(Z; Y)$. Further, given the existence, we present three possible scenarios regarding the observability of target $Y$ and protected attribute $A$. These scenarios include: (1) *Fully-observable* containing both target labels and protected attribute labels, (2) *Partially-observable* lacking target labels but containing protected attribute, and (3) *Non-observable* lacking any labels. We have discussed the first two scenarios. In this section, we mainly discuss the third scenario where universal distribution does not yield any labels. Under this weakest annotation possibility, we seek a simple approach based on self-supervised learning (SSL) to demonstrate the potential of overcoming the trade-off between attribute bias removal and the best attainable performance.

The idea of this approach is to utilize universal distribution to train an encoder that can bring features from the same input closer together while pushing them away from features belonging to other inputs. By doing so, the encoder learns to capture the intrinsic information of the input without being influenced by the data with attribute bias. To this end, we deploy a two-stage training scheme, as shown in Fig. 9, First, we fine-tune the pretrained baseline encoder (Goyal et al., 2022; Mehta et al., 2022) on universal distribution with contrastive loss (Chen et al., 2020). Then, we apply the frozen encoder followed by a shallow regressor trained from scratch in the downstream task to address attribute bias.

Specifically, during the encoder training, given the input $x^U$ from a minibatch of size $N$ in universal distribution, a set of data augmentation techniques randomly augment $x^U$ to be $x_i$ and $x_j$. Following (Chen et al., 2020), the set of augmentation includes RandomResizedCrop, RandomHorizontalFlip, RandomApply(ColorJitter), and RandomGrayscale. Then, a pretrained baseline encoder $E : \mathcal{X} \to \mathcal{Z}$ outputs learned representation $z_i$ and $z_j$, followed by a mapping network $M : \mathcal{Z} \to \mathcal{H}$ to output $h_i$ and $h_j$, respectively. To ensure $z_i$ and $z_j$ are closer to each other than other features, we optimize $E$ and $M$ over the following contrastive loss (Chen et al., 2020) for pair $(i, j)$:

$$\mathcal{L}_{\text{contrastive}}^{E,M}(i, j) = -\log(\frac{\exp(\text{sim}(h_i, h_j)/\tau)}{\sum_{k=1}^{2N} \mathbb{I}(k \neq i) \exp(\text{sim}(h_i, h_k)/\tau)}) \tag{7}$$

where $\text{sim}(h_i, h_j) = \frac{h_i \cdot h_j}{\|h_i\|\|h_j\|}$, $\tau$ is the temperature parameter, and $\mathbb{I}(k \neq i) \in \{0, 1\}$ is an indicator function that equals 1 if and only if $k \neq i$. The loss is computed on both $(i, j)$ and $(j, i)$ for each input $x^U$.

Further, during applying the trained encoder to the biased dataset of downstream task $\{\mathcal{X}^B, \mathcal{Y}\}$, given the input $x^B \in \mathcal{X}^B$, the frozen encoder is used to output learned representation $z$. Then, a regressor $R : \mathcal{Z} \to \mathcal{Y}$ is introduced with the objective:

$$R^* = \arg\min_R \mathcal{L}_{reg}(R(z), y) \tag{8}$$

where $\mathcal{L}_{reg}$ is an appropriate regression loss (L2 loss for continuous attributes and cross-entropy loss for discrete attributes).

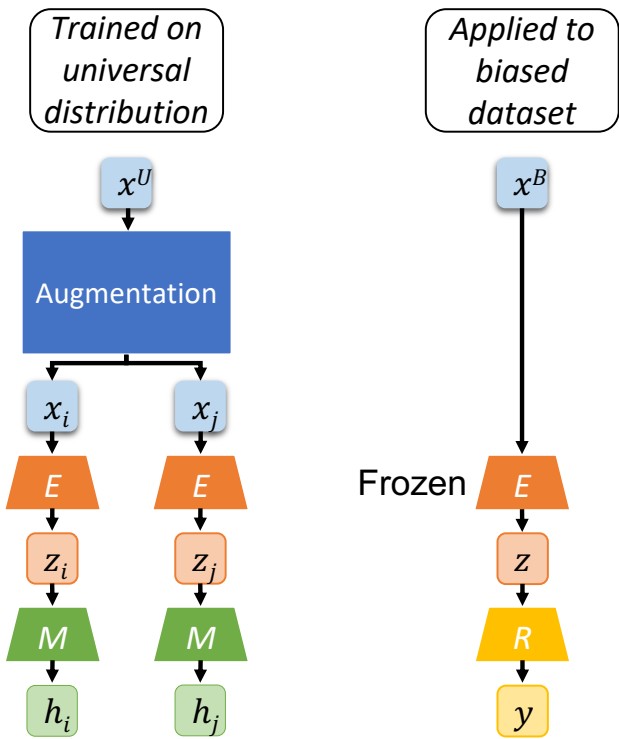

Figure 9: Framework of the proposed approach based on self-supervised learning to utilize universal distribution without any labels. First, the encoder is trained on universal distribution without any labels using contrastive loss. Then, the trained encoder can be applied to the biased dataset for many downstream tasks.

The whole framework can be found in Fig. 9. The network architecture is shown in Tab. 35, and the hyper-parameters are shown in Tab. 31.

**Results.** Following the experiment setup of CelebA in Sec. 6 of the main paper to predict *BlondHair*, we conduct the same experiment on the SSL-based approach. As shown in Tab. 15, compared with its relatively worse performance than baseline without universal distribution, this approach outperforms baseline when getting access to universal distribution, which highlights the possibility to utilize universal distribution to escape the trade-off between attribute bias removal and achieving good accuracy even in the absence of any strengthened supervision signals.

Table 15: Performance of the proposed approach based on self-supervised learning (SSL) under extreme bias in CelebA (*TrainEx* training set) to predict *blond hair*. The SSL-based approach is trained on *TrainEx* appending the dataset shown in parenthesis as universal distribution. Without universal distribution, the performance of this approach is worse than baseline. With universal distribution, its performance is boosted. **Bold** for the best results.

| Method | Test Accuracy | | Mutual Information | |
|---|---|---|---|---|
| | Unbiased ↑ | Bias-conflicting ↑ | $I(Z;A)$ ↓ | $\Delta$ (%) ↑ |
| Baseline | $66.11_{\pm 0.32}$ | $33.89_{\pm 0.45}$ | $0.57_{\pm 0.01}$ | 0.00 |
| SSL | $64.24_{\pm 0.54}$ | $32.59_{\pm 0.61}$ | $0.56_{\pm 0.02}$ | 1.75 |
| SSL (FFHQ) | $69.02_{\pm 0.47}$ | $42.75_{\pm 0.83}$ | $0.51_{\pm 0.02}$ | 10.50 |
| SSL (Synthetic) | $\mathbf{70.19_{\pm 0.58}}$ | $\mathbf{44.23_{\pm 0.92}}$ | $\mathbf{0.50_{\pm 0.02}}$ | **12.28** |

# 8 Conclusion

We mathematically and empirically showed the sensitivity of the state-of-the-art attribute bias removal methods to the bias strength, revealing a previously overlooked limitation of these methods. Specifically, we derived an information-theoretic upper bound on the performance of any attribute bias removal method and verified it in experiments on synthetic, image, and census datasets. These findings caution against the use of existing attribute bias removal methods in datasets with potentially strong bias (*e.g.*, small datasets). Next, we stated a necessary condition for the existence of any method that can remove the extreme attribute bias (*i.e.*, universal distribution). Finally, based on our theoretical analysis, we constructed a new method that can overcome the extreme bias under the necessary condition and outperforms state-of-the-art methods.

**Limitations and Future Directions.** While our method shows promising results, in the ablation studies (Sec. 6.7) we found that it is sensitive to the amount of bias in the universal distribution itself. Thus, an interesting future direction is constructing methods that are less sensitive to the bias in the universal distribution. Another interesting direction is to explore how to construct a universal distribution more efficiently. Also, since the proposed method relies on protected attributes during training, its practical applications with real-world data need to carefully follow data privacy and data collection regulations. Besides, it is important to consider more challenging scenarios where protected attribute labels are absent (Creager et al., 2021; Sohoni et al., 2020) or unknown biases emerge (Li et al., 2022). Finally, it is noteworthy that while the ability to effectively remove protected attributes is valuable, removing them will not always result in a fairer decision, as in some cases rewarding a demographic group might be desirable, a matter discussed more elaborately in (Corbett-Davies & Goel, 2018).

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
