# OpenReview forum: "Removing Strong Attribute Bias from Neural Networks with Adversarial Filtering"
_TMLR — Rejected by TMLR_

### Review · Reviewer_xhCK · 2025-06-24

**Summary Of Contributions:**

This paper proposes an adversarial filtering framework to remove strong attribute bias in neural networks, particularly when the protected attribute is highly predictive of the target label. By formulating and validating an information-theoretic upper bound, the authors highlight the inherent limitations of existing debiasing methods under extreme bias. To address this, they introduce a target-agnostic image filtering method that removes the protected attribute information directly in the input space, enabling improved fairness in downstream tasks.

**Audience:**

Yes

**Claims And Evidence:**

Yes

**Requested Changes:**

see weaknesses

**Strengths And Weaknesses:**

**Strength:**
1. The information-theoretic upper bound is well-motivated and general.
2. The authors evaluate the method on diverse modalities and a broad range of datasets.

**Weakness:**
1. The proposed method follows a common “filter-then-train” paradigm, where protected attribute information is removed as a preprocessing step before downstream training.
2. This approach is conceptually similar to many existing works in fairness, disentanglement, or data sanitization. The paper does not clearly delineate what makes this method novel beyond the use of an adversarially trained encoder-decoder.
3. Since the protected attribute is removed prior to training any downstream model, any filtering error or unintended information loss may propagate into the final task model. The method assumes the filtering process preserves all non-protected features, but there is no formal analysis or quantification of the information loss or distortion introduced by the filter.
4. Many of the baselines used for comparison are from 2019–2021. More recent works in fairness-aware representation learning, causal debiasing, or disentanglement-based methods are not included.
The approach relies on the availability of a universal distribution to train the adversarial filter. However, the practical feasibility of obtaining such data in real-world settings is questionable.
5. The use of L1 loss and adversarial image reconstruction to preserve non-protected features could inadvertently retain residual traces of the protected attribute in the generated images. There is no verification that protected information is fully removed beyond estimating mutual information, nor any adversarial attack-style validation.

---

> ### Author Response · Authors · 2025-08-18
>
> **W1.**
>
> We agree that our method adopts a filter-then-train structure, but it differs from prior approaches in three important ways: (1) it is target-agnostic and does not require downstream task labels, (2) it removes protected attributes directly in the input space rather than relying on task-specific latent representations, and (3) it generalizes across images, tabular, and text data (Tabs. 4–5, 10–11). These distinctions make our method a practical and broadly applicable bias removal tool, going beyond standard preprocessing filters.
>
> **W2.**
>
> We thank the reviewer for raising this point. While our method also adopts an encoder–decoder architecture, its novelty lies in three key aspects: (1) Theoretical contribution: we derive and prove a general information-theoretic upper bound (Theorem 1), establishing the fundamental limitations of bias removal methods — a result absent in prior fairness or disentanglement work; (2) Target-agnostic filtering: unlike prior approaches, our method removes protected attributes without requiring downstream target labels, enabling direct applicability across tasks; (3) Image-based design: our approach operates directly on images rather than representations, making it more readily usable for downstream tasks. Together, these contributions go beyond applying adversarial disentanglement and demonstrate that our method is both theoretically grounded and practically versatile.
>
> **W3.**
>
> We agree that preserving non-protected information is crucial. To this end, we explicitly incorporate reconstruction and adversarial preservation losses (Eqs. 4 and 5) to minimize unintended distortions. More importantly, we empirically evaluate preservation: our method achieves strong performance across 23 non-sex-related downstream attributes in CelebA (Tab. 8), confirming that information beyond the protected attribute is retained. Additionally, Table 13 presents an ablation study analyzing how the strength of the reconstruction loss influences the preservation of other attributes.
>
> **W4.**
>
> Following the experimental setup in Sec. 6, we further compare our method with state-of-the-art approaches, including CAD-VAE [3] and FLAC [4], as shown in the table below. The results indicate that these methods do not adequately address the strong bias problem, which still remains an open challenge, whereas our method is explicitly designed for this setting.
>
> Regarding the universal distribution, we emphasize that our method uniquely operates without requiring target labels, unlike prior approaches. Furthermore, our theoretical results (Theorem 1) demonstrate that some form of less-biased distribution is necessary under extreme bias. In practice, such distributions can be obtained from external corpora (e.g., FFHQ for faces). In the paper, we highlight several practical ways to construct such datasets, making this assumption realistic in real-world applications.
>
> Table A: Area under the curve (AUC) in the strong bias region of CelebA dataset.
> | Method    | Unbiased ↑      | Bias-conflicting ↑|
> |----------|:----------:|:----------:|
> | Baseline    | 24.67±0.72      | 17.18±1.62 |
> | CAD-VAE [4]|   26.12±0.43   |  22.94±0.74    |
> | FLAC [4]| 27.91±0.26    | 24.54±0.48     |
> | Ours    | 28.90±0.94      | 24.61±0.79     |
> | Ours (FFHQ)   | 30.29±0.68     | 25.83±1.00     |
> | Ours (Synthetic)    | 30.20±0.85     | 26.04+1.22     |
>
> [3] Ma, Chenrui, et al. "CAD-VAE: Leveraging Correlation-Aware Latents for Comprehensive Fair Disentanglement." arXiv preprint arXiv:2503.07938 (2025).
>
> [4] Sarridis, Ioannis, et al. "Flac: Fairness-aware representation learning by suppressing attribute-class associations." IEEE Transactions on Pattern Analysis and Machine Intelligence (2024).
>
>
>
> **W5.**
>
> In the paper, we empirically validate attribute removal by training downstream classifiers on the filtered data. For instance, Table 12 shows that sex information is no longer exploitable for prediction when performing sex classification on the filtered data. Meanwhile, Tables 8 and 13 demonstrate that classifiers trained to predict other attributes in CelebA retain strong performance, confirming that non-protected attributes are preserved. We will clarify this point in the paper.

---

> > ### Comment · Reviewer_xhCK · 2025-10-05
> >
> > I would like to thank the authors for their detailed and thoughtful responses to the previous review. While I appreciate the additional experiments and clarifications provided, I find that several of my core concerns remain insufficiently addressed.
> >
> > Specifically, the paper still largely follows the well-established “filter-then-train” paradigm, and the newly emphasized target-agnostic aspect, although interesting, does not appear to substantially change the conceptual foundation of the approach. The theoretical result (Theorem 1) is correctly derived but functions more as a restatement of known information-theoretic trade-offs rather than yielding new actionable insight for algorithm design.
> >
> > Moreover, the empirical validation relies heavily on controlled synthetic settings that may not convincingly demonstrate the robustness or generality of the proposed method in realistic deployment scenarios. The assumption of access to a “universal distribution” is particularly strong and not clearly feasible for most practical applications.

---

> > > ### Author Response · Authors · 2025-10-06
> > >
> > > Thank you for the continued feedback.
> > >
> > > **R1. The Paradigm of our Method.**
> > >
> > > The “filter-then-train” paradigm in our method involves filtering attribute information directly at the original data level (e.g., image) and then training the downstream task model on the resulting filtered dataset. In contrast, existing approaches typically perform bias removal and downstream task prediction simultaneously, removing attribute information at the representation level. Our method’s target-agnostic design and the decoupling of attribute bias removal from downstream tasks enable the filtered dataset to be stored and directly reused by any potential downstream task model or future model for better downstream task performance. Conversely, other methods generate representations that are tightly coupled with their full frameworks, making them unsuitable for direct reuse and requiring complete retraining for each new downstream task.
> > >
> > > **R2. Theoretical Result.**
> > >
> > > To the best of our knowledge, this theorem does not correspond to any previously known information-theoretic trade-off. We abstract each term in the formulation from the attribute bias removal setting. Through theoretical derivation, we establish this theorem. Its intuitive and straightforward nature enables broader generalization. Furthermore, the interpretation of this theorem directly motivates the design of our proposed method. Specifically, it indicates that under extreme bias conditions, constructing a universal dataset is essential; however, assigning target labels across diverse downstream tasks is often impractical. To address this, we introduce a target-agnostic approach that effectively utilizes the universal dataset without relying on target labels.
> > >
> > > **R3. The Experiments under Controlled Synthetic Settings.**
> > >
> > > Besides the controlled synthetic experiments on the Colored MNIST dataset, we also perform more extensive evaluations on the CelebA dataset (image modality) and the Adult dataset (tabular modality). Furthermore, as illustrated in Figure 10, we explore various real-world scenarios that span across vision, tabular, and text domains. The controlled synthetic setting serves primarily as a teaser. In the foundational work that first investigated attribute bias (see Figure 4 in [1]), experiments were conducted on the Colored MNIST dataset but restricted to the moderate bias region, where the correlation between the protected attribute and the target attribute is not strong. In contrast, in Figure 1 of our paper, our study extends this analysis to the strong bias region, which represents a more challenging and practically significant scenario, forming the core motivation of our work.
> > >
> > > [1] Kim, Byungju, et al. "Learning not to learn: Training deep neural networks with biased data." Proceedings of the IEEE/CVF conference on computer vision and pattern recognition. 2019.
> > >
> > > **R4. Assumption of the Universal Distribution.**
> > >
> > > Our method does not rely on or assume access to any additional dataset. The universal dataset is not an integral component of our approach; rather, our method can function independently without it. Our method is uniquely designed to leverage such a universal dataset—specifically, one without target labels—an ability that distinguishes it from existing methods. This becomes particularly valuable under extreme bias conditions, where all approaches inevitably require such a universal dataset to mitigate attribute bias. As shown in Tables 3 and 4, within the strong bias region, our method achieves superior performance even without utilizing a universal dataset. Under extreme bias, Theorem 1 establishes that all methods must depend on the universal dataset to effectively eliminate attribute bias. Existing approaches, however, typically demand access to both protected attribute labels and prediction target labels—an impractical requirement when the number of possible target labels is large. To overcome this limitation and enhance the practicality of the universal dataset, we design our method to operate without the need for target labels. In general, constructing a universal dataset becomes more feasible under our formulation, since collecting target labels is not needed. In practice, our experiments demonstrate that for widely used attribute-bias benchmarks—Colored MNIST, CelebA, and Adult—such a dataset can be readily built using either real-world or synthetic data.

---

### Review · Reviewer_RCsU · 2025-07-16

**Summary Of Contributions:**

This paper studies the setting of extreme attribute bias, that is when the target is perfectly or nearly perfectly associated with a protected attribute such as age/race/sex. The paper demonstrates theoretically and empirically that existing attribute removal methods are ineffective in this setting (unsurprisingly since perfect association cannot be disentangled without additional assumptions). The main additionial assumption of this paper is the existence of a "universal" distribution, where the association is weaker but no labels are available. Unsurprisingly, such additional data allows developing stronger bias removal methods - the paper proposes adversarial filtering as one such method and shows it outperforms baselines that were developed under weaker assumptions. In summary, these findings are not particularly surprising - studying  an unexplored setting (extreme bias but access to low bias unlabeled data), developing a method tailored to this setting (adversarial filtering), and then confirming that this is the strongest method for this particular setting is somewhat self-evident. The relevancy of this depends on whether the unexplored setting is of practical interest - which is unclear (see below)

**Audience:**

Yes

**Broader Impact Concerns:**

I don't have particular broader impact concerns.

**Claims And Evidence:**

Yes

**Requested Changes:**

- a strong baseline compared to training an "adversarial filter" on a universal distribution would be a generative image-to-image model that is prompted to remove the protected attribute (color/sex). While this approach is specific to the image domain, I think it is still helpful to understand what the limits of attribute filtering would be with such a method.
- The paper itself is quite lengthy and the ordering of the sections is somewhat unintuitive to me (Section 3 and Section 6 are more theory-heavy sections but "interrupted" by method and experiments section). I think shortening (moving parts to the appendix) and re-organizing (moving Section 3 and 6 together) would strengthen the presentation of the paper.

Minor:
 - "So decreasing the bound will decrease performance, but increasing the bound will not necessarily improve performance." This sounds contradictory and needs rephrasing - a bound can be tight or vacuous, but this seems symmetric in decreasing /increasing the bound.
 - "To empirically validate Theorem 1 on real-world data" I would expect a theorem does not require empirical validation?

**Strengths And Weaknesses:**

Strength:
 - Removing attribute bias from models is a topic of interest to the community (maybe less so the "extreme" attribute bias studied primarily in this work, but it is still worthwhile studying the extreme ends of what could be relevant)
 - Formalization of the problem in Section 3 is helpful
 - presentation of the method in Section 4 and illustrations in Figure 3 and Figure 4 are well designed and ease the understanding of the method
 - Presentation or experimental design and results is well structured and illustrated

Weaknesses:
 - the assumption of access to a "universal distribution" (low bias but unlabeled) is a strong and in my opinion an unrealistic one. Provided there exist a large corpus with low attribute bias but the labelled training set contains strong (over even perfectly) attribute biased samples, the main question would be: why was such a biased dataset selected for labelling in the first place? some issues are best addressed on the data, not on the method side and the setting studied in this work would fall for me into this class of issues.
 - the results of "adversarial filtering" are not fully convincing: judging from Figure 4 (right), removing the color attribute from digits works well (but Colored MNIST is a toy problem anyway). But removing the sex attribute from CelebA is not convincing to me - granted, the beard is a bit weaker, but the filtered image of the  person is still very prototypically male. Given how much conditional generative models have progressed in recent years, it is hard to believe that this is the best attribute filtering on images that can be accomplished.
 - The presentation is somewhat lengthy and not well organized (see  Requested Changes)

---

> ### Author Response · Authors · 2025-08-18
>
> Thank you for the detailed suggestion. We have updated the paper accordingly and marked the changes in orange.
>
> **W1.**
>
> The universal distribution does not require a perfectly unbiased corpus; it only assumes access to partially observable samples that exhibit weaker bias than the highly skewed training set. Our method can utilize such data even without target labels, which are often naturally available (e.g., external face datasets such as FFHQ). Although addressing bias during data collection is ideal, recollecting large, unbiased labeled datasets is often impractical—particularly when multiple target labels are involved. For instance, CelebA includes many facial attributes, some of which are strongly correlated with sex (e.g., hair color). Expanding the dataset and exhaustively labeling all attributes is infeasible. Therefore, our method complements data-centric solutions by offering a practical approach for situations where only biased labeled data is accessible.
>
> **W2.**
>
> Our goal is not to achieve photorealistic or perceptually perfect attribute editing, but rather to ensure that protected attribute information is effectively removed while preserving other features relevant for downstream tasks. To validate this, we provide both the qualitative visualization in Figure 4 as intuition and the quantitative evidence in Tables 12 and 13. For instance, Table 12 shows that sex classifiers trained on the data filtered by our method perform poorly, confirming that the protected attribute information has been removed.
>
> Although existing generative models (e.g., diffusion-based methods) can perform image editing, they often struggle with prompt instruction-following and hallucination [1,2], which may fail to remove protected attributes and even introduce unintended modifications to other attributes. In contrast, our GAN-based method avoids these issues by producing filtered images where the protected attribute is removed while other information is preserved, as verified by our results.
>
> [1] Li, S., Singh, H., & Grover, A. “Instructany2pix: Flexible visual editing via multimodal instruction following.” Annual Conference of the Nations of the Americas Chapter of the Association for Computational Linguistics (NAACL), 2025.
>
> [2] Kawar, Bahjat, et al. "Imagic: Text-based real image editing with diffusion models." Proceedings of the IEEE/CVF conference on computer vision and pattern recognition. 2023.
>
> **C1.**
>
> We thank the reviewer for the suggestion. To address this, we compare our method with a state-of-the-art text-guided image editing approach. Using images from the CelebA dataset, we provide the prompt: “remove the information about sex in this image while preserving other attributes unchanged.” Following the evaluation settings in Tables 12 and 13, we measure both protected attribute prediction and target attribute prediction. The results show that while the baseline method reduces sex information, it also degrades performance on other attribute classification. In contrast, our method achieves a better balance by reducing sex classification accuracy while preserving the accuracy of other attributes.
>
> | Method | Sex Classification ↓  | Other Attribute Classification ↑ |
> |---------------------|:------------------:|:------------------------------:|
> | Baseline    | 98.25±0.13     | 78.08±0.82    |
> | InstructAny2Pix [1]    | 86.34±0.61   | 72.56±0.94    |
> | Ours  | 84.74±0.38     | 81.62±1.46     |
>
>
> **Minor 1.**
>
> Theorem 1 is formally proven in the Appendix. Empirical results are not necessary for its validity; they are provided to illustrate its implications on real-world datasets. To avoid confusion, we will make this clarification in the caption of Figure 2.

---

> > ### Comment · Reviewer_RCsU · 2025-08-22
> > **Response**
> >
> > I would like to thank the authors for their feedback. My concerns are not fully addressed though - I still believe that the best way of dealing with the issue is in the data collection process and not the modeling part and one should not overstate the cost of labelling data  for the toy datasets studied  in this work. Clearly, large scale data collection can be expensive but this work is not in the large-scale regime (would the proposed method work in the large-scale regime?).
> >
> > In addition, I think it is not appropriate to state that the "protected attribute information has been removed" when it is still clearly visible. Filtering might make it less likely that a downstream classifier uses this filtered attribute for its predictions as a spurious feature but I would expect that one could still train a classifier that recovers the protected attribute from the data with high accuracy (at least for CelebA). It would thus be more appropriate to state that the method reduces the reliance of downstream classifier on the filtered attribute rather than saying that the filtered attribute has been removed.

---

> > > ### Author Response · Authors · 2025-08-24
> > >
> > > Thank you for the continued feedback.
> > >
> > > **R1.**
> > >
> > > We fully agree that data collection practices play a critical role in addressing bias, and we do not claim to replace data-centric solutions. Instead, our work is intended to complement them: in many real-world scenarios, datasets with strong spurious correlations already exist, and reducing their bias through relabeling or recollection is often far more costly than applying our method. In this paper, we demonstrate the effectiveness of our method on CelebA, a dataset widely used in bias-mitigation studies with 202,599 samples, which should not be considered a toy benchmark. Our approach can also generalize beyond CelebA to larger-scale settings (e.g., IMDB Face, as studied in the paper). Since the filter is trained once on partially observable data (i.e., without target labels) and then reused across tasks, it eliminates the need to relabel new data for each downstream application. This allows any face dataset—regardless of size—to be filtered for protected attributes while retaining other attributes for downstream tasks. In contrast, existing methods typically require target labels during training, making them infeasible to scale due to the cost of labeling.
> > >
> > > **R2.**
> > >
> > > Thank you for your suggestion. We will revise the manuscript to avoid overstating our claim and instead emphasize that our contribution lies in reducing the reliance of downstream models on protected attributes. While the qualitative examples (e.g., CelebA faces) may still exhibit some visual cues, we provide further evidence through the sex classification results reported in Table 12. As shown, classifiers trained on the filtered data perform poorly, indicating that the protected attribute is less predictive for downstream models. This suggests that recovering the protected attribute information from the filtered data is difficult.

---

### Review · Reviewer_iSsT · 2025-08-05

**Summary Of Contributions:**

This work addresses the problem of spurious correlations, wherein certain attributes—such as race or gender—are statistically correlated with the task labels in the training dataset. Such correlations can lead the model to rely on these attributes rather than learning the underlying task-relevant features. To tackle this issue, this work introduces an information-theoretic bound, demonstrating that model performance is upper-bounded by both the strength of attribute removal and the degree of attribute bias present in the data. Accordingly, a preprocessing approach is proposed, which involves training a feature generator to disentangle and filter out attribute-related features from the input, followed by the training of a downstream classifier on the filtered features.

**Audience:**

Yes

**Broader Impact Concerns:**

Refer to weakness 3.

**Claims And Evidence:**

Yes

**Requested Changes:**

**Major**

refer to **Strengths And Weaknesses**


**Minor**
1. In Table 18 and 19, “universal distribution is constructed…”, the word “Universal” should be capitalized.
2. It would be good to include a comprehensive table summarizing the results of all methods across all modalities.
3. In Eq. 4, the loss term $L_{\text{pred}}$ also takes a and a’ as inputs; however, this is not reflected in the annotations of Figure 4. Clarifying this would improve readability.
4. For Eq. 2,4,5,6, and 7, it would be helpful to explicitly include the arguments to reduce confusion.

**Strengths And Weaknesses:**

**Strengths**
1. This work explores challenging scenarios characterized by severe spurious correlations, where existing bias removal mechanisms fail to perform effectively.
2. It introduces an information-theoretic framework to model the performance of bias mitigation.
3. The empirical evaluation is comprehensive, covering a wide range of data modalities.
4. The study proposes solutions applicable across various observability scenarios.

**Weaknesses**
1. It would be helpful if the authors could specify whether the universal distribution is only required in cases of extreme bias (i.e., when $H(Y\mid A) = 0$), or whether it is a general requirement for the method.
2. The proposed approach assumes access to an additional dataset (i.e., the universal distribution), which may not be feasible in many practical settings. The authors are encouraged to further discuss the practical implications and limitations arising from this assumption.
3. The method processes the bias attributes, specifically by training a bias attribute classifier (Eq. 3 and 4); however, when such attributes include protected information—such as race or gender—this raises potential ethical concerns and may conflict with legal regulations such as GDPR in certain jurisdictions. The authors should address this issue explicitly in the limitations section.
4. Following up on previous concern, the authors do acknowledge the non-observable scenario in the appendix and propose an SSL approach that does not process the bias attribute. It is recommended that this discussion be expanded and moved into the main text to better highlight the practical relevance and ethical considerations of the proposed method.

---

> ### Author Response · Authors · 2025-08-18
>
> Thank you for your detailed suggestion. We have revised the paper accordingly and highlighted the changes in orange.
>
> **W1.**
>
> By our theoretical proof in Corollary 1 (Necessary Condition), a universal distribution is required only in the case of extreme bias, i.e., when H(Y∣A)=0. Introducing a universal dataset relaxes this condition to $H(Y∣A)>0$. Without such a dataset, no existing method can successfully mitigate attribute bias. In scenarios without extreme bias—for example, when $H(Y∣A)>0$, which we define as the strong bias region—we conducted experiments on Colored MNIST, CelebA, and Adult (see Figure 1, Table 3, and Table 4). Across these experiments, we observe that even without a universal dataset, existing methods can mitigate attribute bias, while our method (denoted as “Ours”) consistently achieves superior performance.
>
> **W2.**
>
> Our method does not assume access to an additional dataset. The universal dataset is not bundled with our approach, and our method can be applied independently without it. Our method is specifically designed to take advantage of a universal dataset without target labels—a capability that other methods cannot leverage—in the extreme bias setting, where all methods necessarily require such a dataset. As shown in Tables 3 and 4, within the strong bias region, we demonstrate that our method, even without a universal dataset, achieves superior performance. In contrast, under extreme bias, a universal dataset is required for all methods to effectively remove attribute bias, as established in Theorem 1. However, existing methods generally rely on both protected attribute labels and prediction target labels, which is impractical given the potentially large variety of target labels. To address this limitation and improve the feasibility of the universal dataset, we design our method to operate without requiring target labels.
>
> **W3.**
>
> We thank the reviewer for raising this point. We have added to the Limitations section that applying our method to real-world data involving protected attributes needs to comply with relevant legal and regulatory requirements (e.g., GDPR).
>
> **W4.**
>
> Thank you for your suggestion. We have incorporated it into the main text.
>
> **C3.**
>
> In Figure 4, $a′$ is positioned to the right of $z$, connected through a circled plus sign (⊕), and serves as input to the right-hand $G_{dec}$ module. Here, $a′$ is symmetrically placed with respect to $a$, while a represents the protected attribute value specific to each input image; $a′$ is set as a constant for all inputs. This design ensures that information about the protected attribute is removed from the generated images.
>
> **C4.**
>
> Thank you for your suggestion. We have adjusted it to ensure alignment and clarity.

---

### Author Response · Authors · 2025-08-18

We sincerely thank the reviewers for their thoughtful feedback and for recognizing the strengths of our work, including the introduction of an information-theoretic framework, the comprehensive empirical evaluation, and the clear presentation of our method. We especially appreciate the acknowledgement that our study addresses challenging scenarios of spurious correlations and explores the extreme attribute bias setting, which is of community interest.

---

### Decision · Action_Editor_pqqD · 2025-11-20

**Recommendation:** Reject

**Additional Comments:**

Two reviewers recommended “learning to accept,” and one reviewer recommended “rejection.”
I recommend rejection, but the authors should be encouraged to resubmit with major revisions.

Additional request when preparing the revision:

**The manuscript needs to be shortened and better organized.**
It is currently quite lengthy, and the ordering of sections is somewhat unintuitive. In the submitted version, Section 3 (“Information-Theoretic Bounds on the Performance of Attribute Bias Removal”) and Section 6 (“Necessary Condition to Remove Extreme Bias”) are theory-heavy sections but are “interrupted” by the method in Section 4 and the experiments in Section 5. Reviewers suggested shortening (moving parts to the appendix) and reorganizing (e.g., grouping Sections 3 and 6 together).
The revised version attempts this but remains lengthy and unintuitive: Section 6 has been moved to Section 4, but a new method section (Section 7, “Approach to Utilize Non-Observable Universal Distribution”) now appears after the Experiments section. I suggest merging Sections 7 and 5 under “Method.”

**Audience:**

Yes

**Audience Explanation:**

At least some members of the TMLR's audience would be interested, as addressing spurious correlations is a significant concern in many modern machine learning applications.

**Claims And Evidence:**

No

**Claims Explanation:**

Some claims require further clarification or removal:

- Reviewer RCsU asked the authors to remove strong or unsubstantiated claims, such as overstating results (“filtered attributes have been removed,” etc.).
- The claim that exploring the region [0, 0.02] of color variance is novel is incorrect. This “strong bias region” has been studied previously, for example in Kehrenberg et al., Null-sampling for interpretable and fair representations (2020). See Fig. 6 of Kehrenberg et al., which corresponds to Fig. 1 in the manuscript; the caption of Fig. 5 in Kehrenberg et al. also refers to “a strongly biased training set.”
Another claim that “[…] existing methods for reducing attribute bias require both target labels and protected attribute labels to utilize any universal distribution” also needs clarification. Kehrenberg et al. used an additional non-spurious dataset without y labels. The proposed method, based on an adversarial objective, also resembles the cVAE method of Kehrenberg et al. and what the manuscript refers to as a “neutral value” corresponds to the “null” value in Kehrenberg et al.
- In the introduction, the authors state:
“In contrast to the state-of-the-art bias-removal methods reviewed in Sec. 2, our method is: 1) target-agnostic (whereas existing methods need both the downstream prediction target and attribute labels to remove bias), 2) removing bias directly in the input space (whereas existing methods try to learn an unbiased latent representation), and 3) a simple data pre-processing for downstream tasks (whereas existing methods need to modify the downstream neural network architecture and its training objective).”
However, prior works such as Kehrenberg et al., 2020; Joo et al., Constructing Fair Latent Space for Intersection of Fairness and Explainability, 2025; Balunović et al., Fair Normalizing Flows, 2022 contradict these claims as, among others, they also remove bias in the input space.
- Reviewers’ concern about the practicality of the extreme-bias setting: “extreme attribute bias when a universal distribution exists” may not be a practically relevant problem. Furthermore, “The requirement of a universal dataset in extreme bias settings-although carefully clarified by the authors-limits the generality of the contribution.” Based on Figure 3 - Illustration of extreme bias and the proposed method, the manuscript claims that the method is broadly useful (“target-agnostic”, applicable to “various downstream tasks”, and designed for realistic scenarios where universal distributions exist without labels).

Based on TMLR acceptance criteria, we recommended “[…] authors to adjust (reduce) their claims”.

**Resubmission Of Major Revision:**

The authors may consider submitting a major revision at a later time.